# projUNN: efficient method for training deep networks with unitary matrices

**Bobak T. Kiani**
MIT
bkiani@mit.edu

**Randall Balestriero**
Meta AI, FAIR
rbalestriero@fb.com

**Yann LeCun**
NYU & Meta AI, FAIR
yann@fb.com

**Seth Lloyd**
MIT & Turing Inc.
slloyd@mit.edu

## Abstract

In learning with recurrent or very deep feed-forward networks, employing unitary matrices in each layer can be very effective at maintaining long-range stability. However, restricting network parameters to be unitary typically comes at the cost of expensive parameterizations or increased training runtime. We propose instead an efficient method based on rank-$k$ updates – or their rank-$k$ approximation – that maintains performance at a nearly optimal training runtime. We introduce two variants of this method, named Direct (projUNN-D) and Tangent (projUNN-T) projected Unitary Neural Networks, that can parameterize full $N$-dimensional unitary or orthogonal matrices with a training runtime scaling as $O(kN^2)$. Our method either projects low-rank gradients onto the closest unitary matrix (projUNN-T) or transports unitary matrices in the direction of the low-rank gradient (projUNN-D). Even in the fastest setting ($k = 1$), projUNN is able to train a model's unitary parameters to reach comparable performances against baseline implementations. In recurrent neural network settings, projUNN closely matches or exceeds benchmarked results from prior unitary neural networks. Finally, we preliminarily explore projUNN in training orthogonal convolutional neural networks, which are currently unable to outperform state of the art models but can potentially enhance stability and robustness at large depth.

## 1 Introduction

Learning in neural networks can often be unstable when networks are very deep or inputs are long sequences of data [5, 83]. For example, vanilla recurrent neural networks (RNNs) have recurrent states that are evolved via repeated application of a linear transformation followed by a pointwise nonlinearity, which can become unstable when eigenvalues of the linear transformation are not of magnitude one. Unitary matrices, which have eigenvalues of magnitude one, can naturally avoid this issue and have been used as a means to overcome these so-called vanishing and exploding gradients [5, 44]. More recently, unitary convolutional layers have been similarly constructed to help build more stable deep networks that are norm-preserving in their transformations [58, 72].

In the RNN setting, prior algorithms to apply $n \times n$ unitary matrices in RNNs have parameterized matrices into layers of unitary or orthogonal transformations or parameterized the Lie algebra of the unitary or orthogonal group (see Table 1). In the layer-wise setting, unitarity is enforced for all values of parameters, but many layers are required to form a composition that can recreate any desired unitary, *i.e.,* fully parameterizing an $n \times n$ unitary requires $O(n)$ layers. By parameterizing the Lie algebra [56, 41], algorithms perform better on common benchmarks but have the drawback

36th Conference on Neural Information Processing Systems (NeurIPS 2022).

Table 1: When training RNNs on inputs with sequence length $T$, PROJUNN achieves nearly optimal runtime complexity while maintaining full parameterization of the unitary manifold.

| Model | Complexity of gradient step | Layers to fully parameterize[a] | Method of parameterization |
|---|---|---|---|
| EURNN (tunable, n layers) [44] | $O(Tn^2)$ | $O(n)$ | Sequence of rotations |
| oRNN (n layers) [66] | $O(Tn^2)$ | $O(n)$ | Sequence of householder reflections |
| full-capacity URNN [82] | $O(Tn^2 + n^3)^b$ | 1 | Parameterized matrix entries |
| expRNN [56] | $O(Tn^2 + n^3)^b$ | 1 | Parameterized matrix in Lie algebra |
| PROJUNN (our method) | $O(Tn^2 + kn^2)^c$ | 1 | Parameterized matrix entries |

[a] layers needed to parameterize the full unitary space, [b] approximations exist which may reduce runtimes though these approximations are not implemented here and can significantly bias the gradient [56], [c] runtime shown for typical setting when $k \ll n$ where $k$ is the rank of gradient updates

that performing gradient optimization on an $n \times n$ unitary requires $O(n^3)$ operations generically per step. Though not an issue with the small to medium sized models used today, this $O(n^3)$ is still $O(n)$ slower than standard methods of forward- and back-propogation in RNNs.

Motivated by the feature that gradients in neural networks are typically approximately low rank, we show that gradient updates to unitary/orthogonal matrices can be efficiently performed in low rank settings. We propose a new model called PROJUNN where matrices are first updated via gradient based optimization and then projected back onto the closest unitary (PROJUNN-D) or transported in the direction of the gradient (PROJUNN-T). PROJUNN has near-optimal runtime complexity unlike other existing algorithms for unitary RNNs (Table 1) and is especially effective even in the most extreme case where gradients are approximated by rank one matrices. In RNN learning tasks, PROJUNN matches or exceeds benchmarks of state-of-the-art unitary neural network algorithms.

Though we present our model first in the RNN setting, we show that there is a direct extension of PROJUNN to the case of orthogonal/unitary convolution which we explore further. Here, we perform unitary/orthogonal convolution in the Fourier domain as inspired by [76]. Our algorithm runs efficiently in the convolutional setting especially for filters of large size and many channels (see Appendix F for more details).

## 2 Related works

Maintaining stability in neural networks via orthogonal or unitary matrices has a rich history of study in machine learning, both from an applied and theoretical perspective. Here, we briefly mention the most related works and algorithms we use in comparison to our PROJUNN. For a more holistic review of prior work in unitary neural networks and other related topics, please see Appendix B.

Unitary neural networks were first designed to address the issue of vanishing and exploding gradients in RNNs while learning information in very long sequences of data more efficiently than existing parameterizations such as the long-short term memory unit (LSTM) [38]. Early algorithms [5, 66] maintained unitarity by constructing a series of parameterized unitary transformations. Perhaps the most effective of these methods is the efficient unitary recurrent neural network (EUNN) [44] which parameterized unitary matrices by composing layers of Givens rotations, Fourier transforms, and other unitary transformations. The unitary RNN (uRNN) of [82] and the Cayley parameterization (scoRNN) of [35] parameterized the full unitary space and maintained unitarity by performing a Cayley transformation. Later, [56] introduced the exponential RNN (expRNN) which parameterized unitary matrices in the Lie algebra of the orthogonal/unitary group. Though the uRNN, scoRNN, and expRNN perform well on benchmarks, their algorithms require matrix inversion or SVD steps which are time-consuming in high dimensions.

For convolutional neural networks, [72] showed how to efficiently calculate the singular values of a linear convolution and proposed an algorithm for projecting convolutions onto an operator-norm ball which relied on a series of costly projection steps. [58] introduced a block convolutional orthogonal parameterization (BCOP) which was faster and more efficient than the methods in [72], but required extra parameters in its parameterization and only parameterized a subset of the space of orthogonal convolutions. Most recently, [73] implemented orthogonal convolutions by parameterizing the Lie algebra of the orthogonal group via their skew orthogonal convolution (SOC)

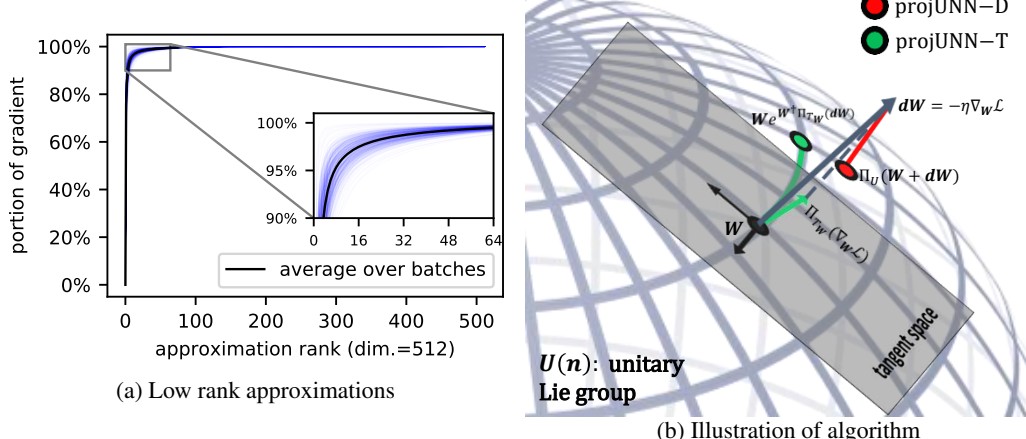

(a) Low rank approximations

(b) Illustration of algorithm

Figure 1: (a) Low rank approximations capture most of the Frobenius norm of the gradient of a $512 \times 512$ matrix in the convolution filter (512 channels) of the last residual block of Resnet-9. Blue lines plot gradients of a single batch during training of our PROJUNN algorithm on CIFAR10 over a single epoch (see Appendix E for details and equivalent plot for RNN architecture). (b) Illustration of a single gradient update via gradient descent with learning rate $\eta$. PROJUNN-D (pictured in red) directly projects the gradient update back onto the unitary/orthogonal manifold. PROJUNN-T (pictured in green) first projects onto the tangent space (Lie algebra) and then performs a rotation in that direction via the exponential map.

algorithm which approximates orthogonal convolutions especially well for small filter sizes. Finally, [76] performs convolutions in the Fourier domain via application of the Cayley transform. Our orthogonal/unitary convolutional parameterization is inspired by their approach and improves their runtime for convolutions over many channels.

## 3 Notation and background

Vectors and matrices are denoted with bold lower-case and upper-case script, $v$ and $V$, respectively. Scalars are denoted by regular script $e$ and tensors are denoted by bold text $\mathbf{T}$. The complex conjugate of a complex-valued input $\cdot$ is denoted by $\cdot^*$ (ignored when real-valued). The transpose of a matrix $M$ is denoted by $M^\intercal$ and the conjugate transpose of a matrix is denoted by $M^\dagger$. We denote the Frobenius norm of a matrix by $\| \cdot \|_F$ and the spectral norm of a matrix by $\| \cdot \|_2$.

Here, we provide a brief overview of the unitary/orthogonal groups and refer readers to Appendix A for a more detailed mathematical background. The set of $n \times n$ orthogonal $O(n)$ and unitary $U(n)$ matrices are both Lie groups defined as

$$O(n) = \left\{ M \in \mathbb{R}^{n \times n} | MM^\intercal = I \right\}, \qquad U(n) = \left\{ M \in \mathbb{C}^{n \times n} | MM^\dagger = I \right\}. \tag{1}$$

Constraining matrices in $O(n)$ and $U(n)$ to have determinant equal to one constructs the special orthogonal $SO(n)$ and unitary $SU(n)$ groups respectively. The Lie algebra or tangent space of the identity of $O(n)$ and $U(n)$ are the set of skew symmetric $\mathfrak{o}(n)$ and skew Hermitian $\mathfrak{u}(n)$ matrices,

$$\mathfrak{o}(n) = \left\{ A \in \mathbb{R}^{n \times n} : A + A^\intercal = 0 \right\}, \qquad \mathfrak{u}(n) = \left\{ A \in \mathbb{C}^{n \times n} : A + A^\dagger = 0 \right\}. \tag{2}$$

The matrix exponential $\exp(\cdot)$ is a map from the Lie algebra to the associated Lie group. The map is surjective if the Lie group is compact and connected – a property which holds for the unitary and special orthogonal groups but not the orthogonal group.

## 4 Projected unitary networks

Our PROJUNN algorithm is motivated by the simple observation that most of the "information" of a typical gradient in a deep learning task is captured in a low rank subspace of the complete gradient.

Figure 15 illustrates this feature when training our PROJUNN convolutional network on CIFAR10. We include further analysis and justification of this low rank behavior in Appendix E. As we will show, we can perform updates on the low rank subspace of the gradient efficiently by approximating the gradient with a low rank matrix and performing projections of parameters onto that low rank subspace. Our experiments show that this methodology, even with rank one approximations, is effective at learning and empirically introduces a form of "beneficial" stochasticity during gradient descent.

Based on how the projection is performed, our PROJUNN algorithm takes two forms illustrated in Figure 1b. The directly projected unitary neural network (PROJUNN-D) projects an update onto the closest unitary/orthogonal matrix in Frobenius norm. The tangent projected unitary neural network (PROJUNN-T) projects gradients onto the tangent space and transports parameters in that direction.

## 4.1 PROJUNN-D

PROJUNN-D takes advantage of the fact that the polar transformation returns the closest unitary or orthogonal matrix in the Frobenius norm to a given matrix (not necessarily unitary or orthogonal):

**Lemma 4.1** (Projection onto unitary manifold [46])**.** *Given a matrix $\boldsymbol{A} \in \mathbb{C}^{n \times n}$:*

$$\Pi_U(\boldsymbol{A}) = \underset{\boldsymbol{U} \in \mathcal{U}(n)}{\arg\min} \|\boldsymbol{A} - \boldsymbol{U}\|_F^2 = \boldsymbol{A}(\boldsymbol{A}^\dagger \boldsymbol{A})^{-\frac{1}{2}}, \tag{3}$$

*where $\mathcal{U}(n)$ indicates the set of $n \times n$ unitary matrices.*

Note, that if the matrix $\boldsymbol{A}$ is real, then the projection above will be onto an orthogonal matrix. Given Lemma 4.1, PROJUNN-D performs optimization in two steps, which are illustrated in Figure 1b. First, matrix entries are updated via a standard learning step as in gradient descent, constructing a new matrix that is generally no longer unitary. In the second step, PROJUNN-D returns the unitary or orthogonal matrix closest in the Frobenius norm to the inputted matrix using Lemma 4.1. At first sight, the second step would require $O(n^3)$ time to perform, but we can take advantage of the fact that gradient updates are typically approximately low rank (see Appendix E). Efficient low rank approximations can be obtained using sampling methods detailed in Section 4.3. With this in mind, we show that rank $k$ updates can be performed in $O(kn^2)$ time when $k \ll n$.

**Theorem 4.2** (Low rank unitary projection)**.** *Let $\boldsymbol{U}$ be an $n \times n$ orthogonal/unitary matrix perturbed by $\boldsymbol{G}_k$, a rank $k$ matrix. Then the projection onto the closest orthogonal/unitary matrix defined below can be performed in $O(k(n^2 + nk + k^2))$ steps.*

$$\boldsymbol{U} + \boldsymbol{G}_k \rightarrow \underset{\boldsymbol{V} \in \mathcal{U}}{\arg\min} \|\boldsymbol{U} + \boldsymbol{G}_k - \boldsymbol{V}\|_F^2. \tag{4}$$

To achieve this runtime, we perform updates completely in an $O(k)$ subspace of the full vector space. The operation $(\boldsymbol{U} + \boldsymbol{G}_k)[(\boldsymbol{U} + \boldsymbol{G}_k)^\dagger (\boldsymbol{U} + \boldsymbol{G}_k)]^{-1/2}$ can be decomposed into a series of $O(k)$ matrix-vector operations and an eigendecomposition of a $2k \times 2k$ sub-matrix. The complete proof and details are deferred to Appendix C. One limitation of the above is that the eigendecomposition and inversion of a low rank matrix can cause numerical instability after many update steps. We discuss this further in Appendix G.3 where we also provide options to alleviate this instability. PROJUNN-T, which we discuss next, does not require matrix inversion and is thus empirically more stable.

## 4.2 PROJUNN-T

PROJUNN-T maintains unitarity of matrices by orthogonally projecting gradient updates onto the tangent space and then performing a rotation in the direction of the projection (*i.e.,* along the geodesic). As in PROJUNN-D, there is a closed form for the orthogonal projection:

**Lemma 4.3** (Tangent space projection [82])**.** *Given the tangent space $T_{\boldsymbol{U}}U(n)$ of an orthogonal/unitary matrix $\boldsymbol{U}$, the orthogonal projection $\Pi_{T_U}$ with respect to the canonical metric $\langle \boldsymbol{X}, \boldsymbol{Y} \rangle = \mathrm{Re}\left(\mathrm{Tr}[\boldsymbol{X}^\dagger \boldsymbol{Y}]\right)$ is*

$$\Pi_{T_U}(\boldsymbol{X}) = \frac{1}{2}\left(\boldsymbol{X} - \boldsymbol{U}\boldsymbol{X}^\dagger \boldsymbol{U}\right). \tag{5}$$

*Similar to Lemma 4.1, this projection also returns the closest matrix in Frobenius norm to $\boldsymbol{X}$ in the tangent space,*

$$\min_{\boldsymbol{Y} \in T_U U(n)} \|\boldsymbol{Y} - \boldsymbol{X}\|_F = \Pi_{T_U}(\boldsymbol{X}). \tag{6}$$

Similar to PROJUNN-D, PROJUNN-T performs learning in two steps. First, a gradient update $\boldsymbol{G}$ is projected onto the tangent space using Lemma 4.3. Then, the orthogonal/unitary matrix is transported or rotated in the direction of the projection by application of the exponential map via the update rule [56, 82],

$$\boldsymbol{U} \rightarrow \boldsymbol{U} \exp\left[-\eta \boldsymbol{U}^\dagger \Pi_{T_U}(\boldsymbol{G})\right], \tag{7}$$

where $\eta$ denotes the learning rate. This update rule is an example of Riemannian gradient descent where we use the exponential map to transport gradient updates along the unitary/orthogonal manifold [15]. Here, we transport the matrix $\boldsymbol{U}$ along the geodesic in the direction of $\Pi_{T_U}(\boldsymbol{G})$. This can be related to the update of PROJUNN-D which is an example of a retraction or an approximation to the exponential map of PROJUNN-T (see Appendix C.3).

The update rule above requires matrix exponentiation and multiplication, both costly steps which can be sped up when $\boldsymbol{G}$ is a low rank matrix. Namely, to perform a rank $k$ gradient update, we obtain an equivalent runtime scaling of $O(kn^2)$ for the PROJUNN-D when $k \ll n$.

**Theorem 4.4** (Low rank tangent transport). *Let $\boldsymbol{U}$ be an $n \times n$ orthogonal/unitary matrix perturbed by $\boldsymbol{G}_k$, a rank $k$ matrix. Then projecting $\boldsymbol{G}_k$ onto the tangent space and performing a rotation in that direction as defined in Equation (7) can be performed in $O(k(n^2 + nk + k^2))$ steps.*

As with the PROJUNN-D, we achieve this runtime by performing the update above completely in an $O(k)$ subspace of the full vector space. The update via the exponential map can similarly be decomposed into a series of $O(k)$ matrix-vector operations and an eigendecomposition of a $2k \times 2k$ sub-matrix. Proper manipulations of the eigenvalues of the sub-matrix implement updates via the exponential map. The complete proof and details are deferred to Appendix C.

### 4.3 Sampling methods

Commonly, gradients can have large rank but have still have many small singular values (*e.g.,* see Figure 1a). Here, a matrix $\boldsymbol{A}$ is deemed approximately low rank (see more details in Appendix E), and one can obtain a rank $k$ approximation $\boldsymbol{A}_k$ of $\boldsymbol{A}$ by sampling from rows and columns of $\boldsymbol{A}$. We use two sampling algorithms. The **LSI sampling** algorithm [68] obtains a rank $k$ approximation to an $n \times n$ matrix $\boldsymbol{A}$ in time $O(kn^2 \log n)$. The algorithm projects the matrix $\boldsymbol{A}$ onto a random orthogonal subspace and then applies SVD based methods to the projected matrix to obtain the low rank approximation to that matrix. This algorithm features low approximation errors even for small $k$ and is used extensively in our implementation. The **column sampling** (linear time SVD) algorithm [23] samples from the columns of an $n \times n$ matrix $\boldsymbol{A}$ to obtain a rank $k$ approximation in $O(c^2 n + c^3)$ time, where $c$ is a hyperparameter indicating the number of columns sampled. Typically, $c$ is chosen as a multiple of $k$ so the runtime is $O(k^2 n + k^3)$. In implementing this algorithm, we calculate the right singular vectors via matrix multiplication of the left singular vectors so the total runtime is $O(kn^2 + k^2 n + k^3)$.

We note that the two procedures described above, though sufficient for our purposes, can be further optimized in their asymptotic runtime. For sake of completeness, we discuss two of these other sampling algorithms in Appendix E.

### 4.4 Extension to unitary or orthogonal convolution

Unitary/orthogonal convolutions are linear convolution operations that also preserve the 2-norm (isometric). Restricting convolutions to be unitary/orthogonal typically results in a drop in performance on standard imaging tasks when used in isolation, but prior work has explored unitary/orthogonal convolutions to potentially improve algorithmic stability and robustness (see Appendix B.1 for more background) [58, 76]. We describe here how PROJUNN can be used to implement unitary/orthogonal convolutions in potentially a more efficient manner.

Given input tensor $\mathbf{X} \in \mathbb{C}^{M \times N \times C}$ where $C$ is the number of channels of an $M \times N$ input, linear convolution (or technically cross-correlation) with a filter $\mathbf{W} \in \mathbb{C}^{M \times N \times C \times C}$ is defined as

$$[\text{conv}_{\mathbf{W}}(\mathbf{X})]_{p,q,d} = \sum_{c=1}^{C} \sum_{m=1}^{M} \sum_{n=1}^{N} \mathbf{W}_{m,n,d,c} \mathbf{X}_{p+m,q+n,c}, \tag{8}$$

where the indexing above is assumed to be cyclic (taken modulus the corresponding dimension) [55, 28]. Orthogonal/unitary convolutions form a subset of filters that preserve norms, *i.e.,* filters $\mathbf{W}$

such that $\|\operatorname{conv}_{\mathbf{W}}(\mathbf{X})\| = \|\mathbf{X}\|$. Equivalently, $\operatorname{conv}_{\mathbf{W}}(\cdot)$ is orthogonal/unitary if the Jacobian of the transformation is also orthogonal/unitary. To maintain unitarity/orthogonality, we set the dimensions of the filter $\mathbf{W}$ above such that it returns an output $\mathbf{Y}$ of the same dimension as the input $\mathbf{X}$. One can also perform semi-orthogonal or semi-unitary convolution by appropriately zero-padding an input or truncating from dimensions in the output.

Standard convolutional filters are typically supported over a sparse set of local elements, but performing orthogonal/unitary convolution generally requires implementing convolutions with filters supported over all elements resulting in slower runtimes. One can locally parameterize convolutional filters in the Lie algebra of the orthogonal/unitary group; nevertheless the exponential map into the Lie group expands the support of the filter:

$$\exp[\operatorname{conv}_{\mathbf{L}}](\mathbf{X}) = \mathbf{X} + \mathbf{L} * \mathbf{X} + \frac{1}{2}\mathbf{L} *^2 \mathbf{X} + \frac{1}{6}\mathbf{L} *^3 \mathbf{X} + \cdots \tag{9}$$

Thus, enforcing unitarity in convolutions generally requires additional overhead over the traditional setting of locally supported filters, but by performing convolution in the Fourier domain, runtimes for full-width filters can be optimally improved to $O(N^2 C \log(N) + N^2 C^2)$ [64]:

$$[\operatorname{FFT} \operatorname{conv}_{\mathbf{W}}(\mathbf{X})]_{\widehat{r},\widehat{s},:} = \widehat{\mathbf{W}}_{\widehat{r},\widehat{s},:,:}^{*} \ [\operatorname{FFT} \mathbf{X}]_{\widehat{r},\widehat{s},:} \ , \tag{10}$$

where $\widehat{\mathbf{W}}_{i,j,:,:}$ is the value of the $\widehat{r}$ and $\widehat{s}$ frequency of $\mathbf{W}$ across all channels in the Fourier domain and FFT is the 2-dimensional fast Fourier transformation.

Our method is inspired by that of [76] which transformed $\mathbf{W}$ into Fourier space and performed a Cayley transformation (approximation to the exponential map into the Lie group) over the matrices indexed by $\widehat{\mathbf{W}}_{\widehat{r},\widehat{s},:,:}$ which requires $O(N^2 C^2 \log(N) + N^2 C^3)$ operations. For our algorithm, we parameterize $\mathbf{W}$ in the Fourier domain and only manipulate $\widehat{\mathbf{W}}$ (see Appendix B.1 for a depiction of our parameterization). By parameterizing $\widehat{\mathbf{W}}$ directly and performing rank $k$ updates using our PROJUNN, this runtime can be improved to $O(N^2 C \log(N) + k N^2 C^2)$ which is optimal when $k \ll N$. Our procedure for performing unitary/orthogonal convolution on an input $\mathbf{X}$ with filter $\mathbf{W}$ essentially follows the steps in Equation (10): perform an FFT on $\mathbf{X}$, block-multiply this by $\widehat{\mathbf{W}}$, and perform an inverse FFT on the output to obtain the final result.

**Limitations**  Unitary/orthogonal convolutions are implemented in a cyclic fashion (*i.e.,* indices are taken modulus the dimension) which is not the standard approach but has been used before to accelerate convolutional operations [64]. Additionally, we parameterize convolution filters to have support over all possible elements (full-width), which can be expensive in memory. One can restrict the convolution to local terms in the Lie algebra, but this would not improve runtime as our algorithm runs in the Fourier space. To target local terms in a convolution, we instead propose for future work to implement a regularizer which has a specified support and penalizes the norm of the filter outside that support. Finally, the space of orthogonal convolutions has multiple disconnected components, which can present challenges for gradient based learning [58]. However, we can avoid this drawback by implementing PROJUNN using fully supported filters in the space of unitary convolutions which is connected (proof deferred to Appendix C.4).

**Theorem 4.5** (Unitary convolutional manifold is connected). *The space of unitary convolutions with filters of full support has a single connected component.*

### 4.5   Pseudocode for performing projUNN updates

Pseudocode for performing an update step on a unitary or orthogonal matrix $U$ with a gradient update of $\Delta U$ is shown in Algorithm 1. In convolutional settings, the steps in Algorithm 1 are applied across blocks of the convolution in Fourier space which can be performed in parallel. As a cautionary note, especially in the last step of Algorithm 1, where there is a composition of multiple matrix-vector multiplications, the order of these multiplications must be chosen to only perform matrix-vector operations to ensure optimal runtime. In other words, two $N \times N$ matrices should never be multiplied by each other at any point in this algorithm.

**Algorithm 1** PROJUNN update step

**Require:** unitary matrix $\boldsymbol{U} \in \mathbb{C}^{N \times N}$ or orthogonal matrix $\boldsymbol{U} \in \mathbb{R}^{N \times N}$
**Require:** gradient update $\Delta \boldsymbol{U} \in \mathbb{C}^{N \times N}$ or $\Delta \boldsymbol{U} \in \mathbb{R}^{N \times N}$
**Require:** hyperparameter $k$ corresponding to rank of approximation
1: Obtain rank $k$ approximation to $\Delta \boldsymbol{U}$ with output $\sum_{i=1}^{k} \boldsymbol{a}_i \boldsymbol{b}_i^{\dagger} \approx \Delta \boldsymbol{U}$ (see Section 4.3)
2: Follow steps in Theorem 4.2 (PROJUNN-D) or Theorem 4.4 (PROJUNN-T) in Appendix C:
3:  Perform Gram-Schmidt (via QR decomposition) on concatenation of vectors $\boldsymbol{U}^{\dagger} \boldsymbol{a}_i$ and $\boldsymbol{b}_i$ for all $i \in [k]$:
    output $\boldsymbol{Q} \in \mathbb{C}^{N \times k}$ as semi-orthogonal matrix containing basis after Gram-Schmidt
4:  Form matrix $\boldsymbol{K} \in \mathbb{C}^{2k \times 2k}$ below:
    PROJUNN-D: $\boldsymbol{K} = \sum_{i=1}^{k} \boldsymbol{Q}^{\dagger} \boldsymbol{U}^{\dagger} \boldsymbol{a}_i \boldsymbol{b}_i^{\dagger} \boldsymbol{Q} + \boldsymbol{Q}^{\dagger} \boldsymbol{b}_i \boldsymbol{a}_i^{\dagger} \boldsymbol{U} \boldsymbol{Q} + \sum_{i=1}^{k} \sum_{j=1}^{k} (\boldsymbol{a}_i^{\dagger} \boldsymbol{a}_j) \boldsymbol{Q}^{\dagger} \boldsymbol{b}_i \boldsymbol{b}_j^{\dagger} \boldsymbol{Q}$
     *see Equation* (C.1) *to Equation* (C.6)
    PROJUNN-T: $\boldsymbol{K} = \frac{1}{2} \left[ \sum_{i=1}^{k} \boldsymbol{Q}^{\dagger} \boldsymbol{U}^{\dagger} \boldsymbol{a}_i \boldsymbol{b}_i^{\dagger} \boldsymbol{Q} - \boldsymbol{Q}^{\dagger} \boldsymbol{b}_i \boldsymbol{a}_i^{\dagger} \boldsymbol{U} \boldsymbol{Q} \right]$
     *see Equation* (C.12) *to Equation* (C.15)
5:  Find eigenvalues $s_1, \ldots, s_{2k}$ and eigenvectors $\boldsymbol{u}_1, \ldots, \boldsymbol{u}_{2k}$ of $\boldsymbol{K}$
6:  Perform update step by applying eigenvalue function:
    PROJUNN-D: $\boldsymbol{U} \leftarrow (\boldsymbol{U} + \sum_{i=1}^{k} \boldsymbol{a}_i \boldsymbol{b}_i^{\dagger}) \left[ \boldsymbol{I} + \sum_{j=1}^{2k} \left( (s_j + 1 + \epsilon)^{-\frac{1}{2}} - 1 \right) \boldsymbol{u}_j \boldsymbol{u}_j^{\dagger} \right]$
     *see Equation* (C.8) *and Equation* (C.9), $\epsilon$ added for stability when $s_j \approx -1$ (we set $\epsilon = 10^{-8}$)
    PROJUNN-T: $\boldsymbol{U} \leftarrow \boldsymbol{U} \left[ \boldsymbol{I} + \sum_{j=1}^{2k} (\exp(-\eta s_j) - 1) \boldsymbol{u}_j \boldsymbol{u}_j^{\dagger} \right]$ where $\eta$ is the learning rate
     *see Equation* (C.17) *and Equation* (C.18)

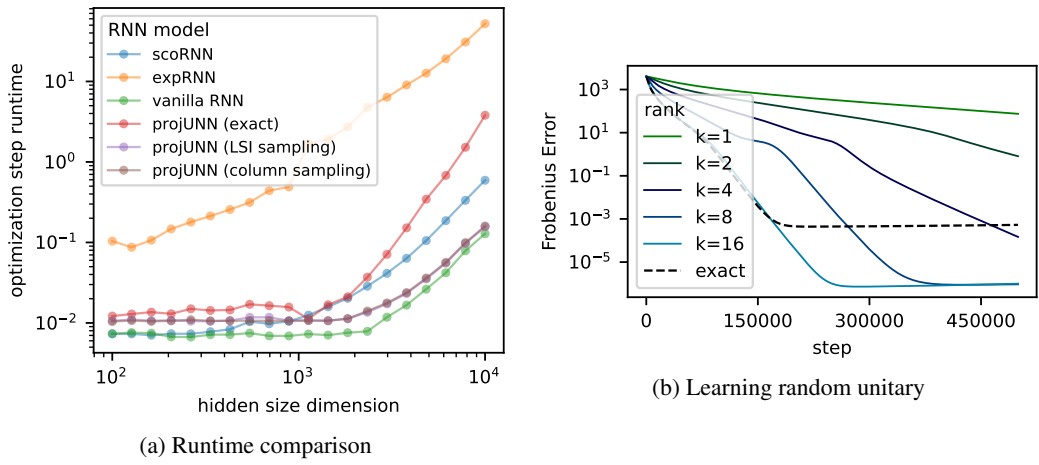

(a) Runtime comparison

(b) Learning random unitary

Figure 2: (a) Runtime of PROJUNN (with low rank approximation) scales asymptotically at same rate of a vanilla RNN and much faster than other unitary RNN models or the exact version of PROJUNN (not using low rank approximation). Practical runtime improvements are achieved when the hidden dimension is large (see Appendix F for details). (b) PROJUNN-T can learn a random target unitary matrix using SGD. For a fixed learning rate, the loss decays at a rate proportional to the approximation rank $k$ up to $k = 16$ where the approximation captures the full batch size (see exact PROJUNN which employs no approximation). The y-axis plots Frobenius error $\|\boldsymbol{U} - \boldsymbol{U}_{tar}\|_F^2$.

## 4.6 Runtime comparisons

PROJUNN has a nearly optimal asymptotic runtime scaling which offers practical benefits in high dimensions. In the RNN setting, Figure 2a shows that the low rank version of PROJUNN has a runtime that scales at the same rate as that of a vanilla RNN albeit with increased overhead. Updating the unitary matrix of PROJUNN takes $O(kn^2)$ time for performing updates of rank $k \ll n$, only a factor $k$ more than a vanilla RNN which performs updates in $O(n^2)$ time. Note, that exact (full rank) updates to the $n \times n$ unitary matrices of a PROJUNN take roughly $O(n^3)$ time corresponding to the runtime of an SVD and equivalent to the runtime of expRNN and scoRNN [56, 35].

In the convolutional setting, PROJUNN offers the most benefit when there are many channels, filters with large support (very wide), or a need for exact unitary/orthogonal operations (in contrast with an

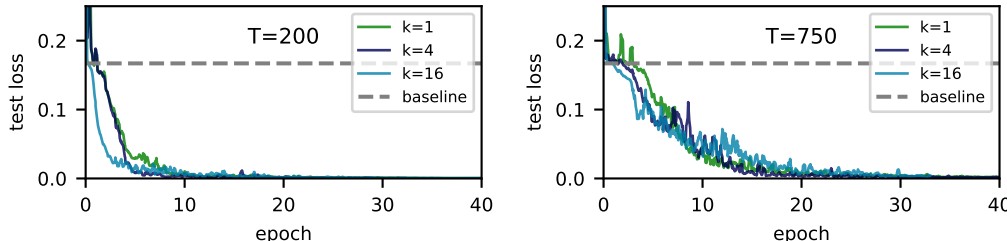

Figure 3: PROJUNN-T learns the adding task with $T = 200$ and $T = 750$. Test error is smoothed by taking the running average of 5 sequential points. See Appendix D.2 for more details.

approximate method like [73]). Given an $N \times N$ input with $C$ channels, a forward and backward pass of PROJUNN runs in time $O(N^2 C \log(N) + kN^2C^2)$ when performing rank $k$ updates. This is a factor of $C$ faster than the Cayley implementation [76] which runs in time $O(N^2C^2\log(N) + N^2C^3)$. For a more complete analysis of the asymptotic and empirical runtimes of various models including many not listed here, please see Appendix F.

## 5 Experiments

We propose in this section a variety of benchmarked experiments to validate the efficiency and performance of the proposed PROJUNN method focusing mostly on RNN tasks.[1] We include further details of the experiments in Appendix D including a preliminary empirical analysis of PROJUNN in convolutional tasks.

**Toy model: learning random unitary**  To study the learning trajectories of PROJUNN, we consider a simple toy model aimed at learning a target random unitary. More specifically, we parameterize a large unitary matrix $U \in \mathbb{C}^{2048 \times 2048}$ to learn a Haar random target unitary $U_{tar} \in \mathbb{C}^{2048 \times 2048}$ given a dataset $\{x_i, y_i = U_{tar}x_i\}_{i=1}^{4096}$ of size 4096 where $x_i \in \mathbb{C}^{2048}$ has entries drawn i.i.d. random normal. $U$ is initialized as a random unitary matrix, and each step, we perform vanilla gradient descent over a batch of 16 training points using mean-squared error loss $\ell(x_i, y_i) = \|Ux_i - y_i\|_2^2$. Approximations of rank $k$ to the gradient are obtained using the column sampling algorithm.

Figure 2b, which plots the Frobenius error $\|U - U_{tar}\|_F^2$, shows that PROJUNN-T equipped with the column sampling approximator is able to learn the random target unitary even when $k = 1$ (see Appendix D.1 for plots with PROJUNN-D). Furthermore, for a fixed learning rate, learning requires fewer steps with larger $k$ up to $k = 16$, the maximum rank of the gradient (note that $\nabla_U \ell(x_i, y_i)$ is rank 1). Therefore, approximating the gradient via low rank approximations can significantly speed up learning in this task (see Appendix D.1 for further details).

**Adding task**  In the adding task, an RNN must learn to add two numbers in a long sequence. We consider a variant of the adding task studied in [5], where the input consists of two data sequences of length $T$. The first is a list of $T$ numbers sampled uniformly from $[0, 1]$, and the second is a list of binary digits set to zero except for two locations (those which must be summed) set to one located uniformly at random within the intervals $[1, T/2)$ and $[T/2, T)$ respectively.

Consistent with [35], we train our PROJUNN-T using an RNN with hidden dimension of 170 and the RMSprop optimizer to reduce the mean-squared error of the output with respect to the target. Naively predicting the average value of one for a random input achieves mean-squared error of approximately 0.167. As shown in Figure 3, PROJUNN-T is able to learn the target function even with rank $k = 1$ approximations. Surprisingly, for a fixed learning rate and scheduler, convergence to the true solution is almost equally fast for $k = 1$, $k = 4$, and $k = 16$. Further details are provided in Appendix D.2.

**Copy memory task**  The copying memory task is a common benchmark for RNNs [38, 5, 36], where the aim is to memorize input data by ignoring a long sequence of void tokens. Given an

---

[1]code repository: https://github.com/facebookresearch/projUNN

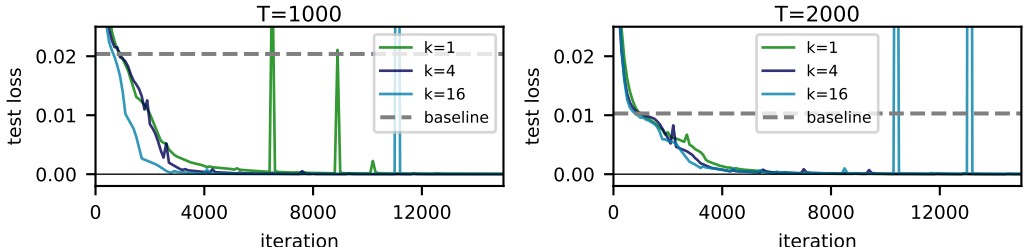

Figure 4: PROJUNN-T equipped with the column sampling approximation learns the copy task with $T = 1000$ and $T = 2000$ even with rank one approximations.

Table 2: Result of gradient descent optimization using the RMSprop optimizer on a single layer RNN for the permutedMNIST classification task. Each result is averaged over 3 runs, the same cross validation is done for all settings and includes the learning rate and its schedule. Training occurs for 200 epochs, and $10\%$ of the training set (same for all models) is set apart as validation set. The training curves are provided in Figure 13.

| | | | | | | | | PROJUNN-D | | | | | PROJUNN-T | | | | |
|---|---|---|---|---|---|---|---|---|---|---|---|---|---|---|---|---|---|
| Width | RGD | LSTM | ScoRNN | ExpRNN | $DT_\infty$ | $DT_{100}$ | $DT_1$ | k=1 | 2 | 4 | 8 | 16 | k=1 | 2 | 4 | 8 | 16 |
| 116 | 92.5 | 91.8 | - | - | - | - | - | 92.8 | 93.0 | 93.0 | 92.9 | **93.2** | 92.5 | 92.6 | 92.5 | 93.0 | 92.8 |
| 170 | - | 92.0 | 94.8 | 94.9 | 95.0 | 95.1 | **95.2** | 94.3 | 94.3 | 94.4 | 94.7 | 94.3 | 94.4 | 94.3 | 94.4 | 94.1 | 94.3 |
| 360 | 93.9 | 92.9 | 96.2 | 96.2 | **96.5** | 96.4 | 96.3 | 96.4 | 96.4 | 96.3 | 96.3 | **96.5** | 96.3 | 96.3 | 96.4 | 96.2 | 96.4 |
| 512 | 94.7 | 92.0 | 96.6 | 96.6 | 96.8 | 96.7 | 96.7 | **97.0** | **97.0** | 96.8 | 96.9 | **97.0** | 96.7 | 96.7 | 96.8 | 96.8 | 96.7 |

alphabet of $n + 2$ symbols $\{a_i\}_{i=1}^{n+2}$, $n$ of which represent data (sequence of letters $A, B, \dots$) and additional *void* (-) and *start recall* (:) tokens, the RNN must output the first $K$ input tokens as the last $K$ output tokens and *void* otherwise. An example input/output for $M = 6$ with $n = 4$ is

```
Input:   ABCDAD---···--------:-----
Output:  ------------···-------ABCDAD
```

Here, $T = 1000$ or $T = 2000$ so the network must memorize data over a very long sequence of void tokens. As in [44], we consider $n = 8$ and input length $K = 10$ and train networks with batch size 128 using the RMSProp algorithm. Naively predicting $T + K$ void tokens followed by $K$ random selections of the $n$ possible tokens achieves a baseline loss of $K \log(n)/(T + 2K)$. PROJUNN-T is able to learn the copy task efficiently as shown in Figure 4. In fact, for fixed learning rates, rank one approximations using the column sampling algorithm provide the fastest convergence to the true solution in comparison to higher rank approximations. Networks were initialized using Henaff initialization (see Appendix G.4) and the learning rate for unitary parameters was set to 32 times less than that of regular parameters (see Appendix D.3 for more details).

**Permuted MNIST**  Another challenging long-term memory task we consider is the permuted pixel-by-pixel MNIST dataset. Here, MNIST images are flattened, and pixels are randomly shuffled and placed in a sequence thereby creating some non-local dependencies. MNIST images have $28 \times 28$ resolution, so the pixel-by-pixel sequences have length $T = 784$. The task is digit classification (10 classes) as in standard MNIST models. We employ the same data processing, shuffle permutation, and formatting as that in prior works [56]. We perform cross-validation over different learning rates and evaluate both PROJUNN-T and PROJUNN-D with different low-rank values $k \in \{1, 2, 4, 8, 16\}$. The final test accuracy is shown in Table 2. As observed in the copy and adding tasks, we find that using $k > 1$ does not lead to improved performances. In fact, we provide the evolution of the test set accuracy during training in Figure 13 and note that as the number of updates is large (hundreds per epoch), even rank $k = 1$ update are able to move the model's parameters to their local optimum.

**CNN experiments**  To explore the performance of our PROJUNN training algorithm for convolutional layers, we first analyzed its performance on CIFAR10 classification using a Resnet architecture [34]. Our aim was not to "beat" benchmarks but to provide an honest comparison of the performance of PROJUNN to existing methods. In fact, as noted earlier, enforcing unitarity generically results in a drop in accuracy for commonly used architectures. Consistent with prior work [76] we employ

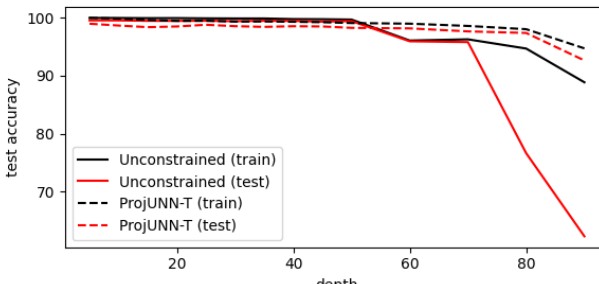

Figure 5: PROJUNN can more stably train very deep CNNs. Training on MNIST is done for 50 epochs in all cases with conv2d-BN-ReLU blocks (repeated "depth" times) and learning rate cross-validation (RMSprop), 32 channels throughout, and a final linear classifier. For 100 epochs and a depth of 100, we obtain $92.7, 23.5$ for the train/test accuracy of unconstrained CNN, and $95.7, 94.6$ for projUNN-T.

data-augmentation of random translations and left-right flips. Previous analysis in the RNN setting showed that rank $k = 1$ is sufficient for convergence so we always set $k = 1$ when using PROJUNN in the convolutional setting. For Resnet9 trained using the RMSprop optimizer, PROJUNN-T and PROJUNN-D reached $80.75\%$ and $82.06\%$ accuracy respectively, matching or outperforming reported results from existing unitary CNN models which achieved accuracies of $80.72\%$ for BCOP [58] and $81.70\%$ for Cayley [76] (further details in Appendix D.5). Note, that all of these methods resulted in a performance drop compared to the standard model (without unitary constraints) which achieved accuracy of $92.26\%$. Hence, we believe that there remain a large potential for unitary models to close this gap. Separate from just performance and to motivate the use of unitary parameterization, we provide in Figure 5, test accuracy results from a simple CNN model with progressively increasing depth trained with and without unitary parameterization on MNIST data. We observe that unitary weights might provide benefits for vanilla CNN architectures that have not been designed to handle very deep settings. Of course, various techniques and tricks have been designed to enable CNNs to be trainable at large depths [83, 34, 14]. Unitary convolutions, which are simple and theoretically motivated, can potentially be used either separately or in-tandem with these other techniques.

## 6 Discussion

Our PROJUNN shows that one need not sacrifice performance or runtime in training unitary neural network architectures. Our results broadly take advantage of the approximate low rank structure of parameter gradients to perform updates at nearly optimal runtime. Looking beyond the setting studied here, it is an interesting question how our framework can be applied to other neural network architectures or parameter manifolds. Group convolutional neural networks and Riemannian gradient descent offer two promising avenues for further application of our techniques.

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
