# A Mathematical background

## A.1 Lie groups and Lie algebras

Here, we provide a brief mathematical background to Lie groups and Lie algebras with a particular focus on the unitary and orthogonal groups. For a more comprehensive overview, we recommend [32].

Though Lie groups are typically defined with respect to differentiable manifolds, we restrict ourselves here to the subset of matrix Lie groups which is less general but allows for a more concise and simpler theoretical overview. Informally, Lie groups are groups whose elements are specified by a set of parameters that vary smoothly and continuously, *i.e.,* the group is also a differentiable manifold. Specific to matrices, Lie groups are commonly defined with respect to the general linear group $GL(n, \mathbb{C})$ denoting the set of invertible $n \times n$ matrices with complex valued entries [32]. Lie groups are closed subgroups of $GL(n, \mathbb{C})$ that have the following smoothness property.

**Definition A.1** (Matrix Lie groups [32]). A matrix Lie group is any subgroup of $GL(n, \mathbb{C})$ with the property that any sequence of matrices $\boldsymbol{A}_m \in \mathbb{C}^{n \times n}$ in the subgroup converge to a matrix $\boldsymbol{A}$ that is either an element of the subgroup or not invertible (*i.e.,* not in $GL(n, \mathbb{C})$).

Two important Lie groups studied in this work are the unitary and orthogonal groups whose definitions are copied below.

$$O(n) = \left\{ \boldsymbol{M} \in \mathbb{R}^{n \times n} | \boldsymbol{M}\boldsymbol{M}^\intercal = \boldsymbol{M}^\intercal\boldsymbol{M} = \boldsymbol{I} \right\}, \tag{A.1}$$

$$U(n) = \left\{ \boldsymbol{M} \in \mathbb{C}^{n \times n} | \boldsymbol{M}\boldsymbol{M}^\dagger = \boldsymbol{M}^\dagger\boldsymbol{M} = \boldsymbol{I} \right\}. \tag{A.2}$$

The Lie algebra is the tangent space of a Lie group at the identity element. To observe this, we introduce the matrix exponential map which is central to the connection between Lie groups and their corresponding Lie algebras.

$$\exp(\boldsymbol{M}) = \sum_{k=0}^{\infty} \frac{\boldsymbol{M}^k}{k!}. \tag{A.3}$$

For compact groups, the exponential map is a smooth map whose image is the connected component to the identity of the Lie group [52, 32]. The special orthogonal and unitary groups are both compact and connected so the exponential map is surjective for these groups (*i.e.,* for every group element, there exists an element of the Lie algebra whose exponential is equal to that group element). However, the orthogonal group has two connected components, *i.e.,* elements with positive and negative determinant, and the image of exponential map are only orthogonal matrices with positive determinant.

Since the matrix exponential map is a smooth function, we can take the derivative of the exponential map with respect to a parameter as below.

$$\frac{d}{dt} \exp(t\boldsymbol{X}) = \boldsymbol{X} \exp(t\boldsymbol{X}) = \exp(t\boldsymbol{X})\boldsymbol{X}, \tag{A.4}$$

and thus,

$$\frac{d}{dt} \exp(t\boldsymbol{X})\Big|_{t=0} = \boldsymbol{X}. \tag{A.5}$$

The above gives us the Lie algebra to a given group.

**Definition A.2** (Lie algebra [32]). Given a Lie group $G$, the Lie algebra $\mathfrak{g}$ of $G$ is the set of matrices $\boldsymbol{X}$ such that $e^{t\boldsymbol{X}} \in G$ for all $t \in \mathbb{R}$.

Typically, Lie algebras of a Lie group are denoted with Gothic or Fraktur font. Using the above definition, one can construct the corresponding Lie algebras. As an example, consider the unitary group where given a matrix $\boldsymbol{U} \in U(n)$ and $\boldsymbol{X} \in \mathfrak{u}(n)$,

$$\boldsymbol{U}^{-1} = \boldsymbol{U}^\dagger \iff \exp(t\boldsymbol{X})^{-1} = \exp(-t\boldsymbol{X}) = \exp(t\boldsymbol{X})^\dagger = \exp(t\boldsymbol{X}^\dagger), \tag{A.6}$$

and since the above holds for all $t \in \mathbb{R}$, we can differentiate the above at $t = 0$, obtaining the property of elements of the unitary Lie algebra that $-\boldsymbol{X} = \boldsymbol{X}^\dagger$ as seen in the main text:

$$\frac{d}{dt} \exp(-t\boldsymbol{X})\Big|_{t=0} = \frac{d}{dt} \exp(t\boldsymbol{X}^\dagger)\Big|_{t=0} \tag{A.7}$$
$$-\boldsymbol{X} = \boldsymbol{X}^\dagger.$$

Proceeding in a similar fashion with the orthogonal group, we obtain the following result copied from the main text,

$$\mathfrak{o}(n) = \left\{ \boldsymbol{A} \in \mathbb{R}^{n \times n} : \boldsymbol{A} + \boldsymbol{A}^\intercal = 0 \right\}, \tag{A.8}$$

$$\mathfrak{u}(n) = \left\{ \boldsymbol{A} \in \mathbb{C}^{n \times n} : \boldsymbol{A} + \boldsymbol{A}^\dagger = 0 \right\}. \tag{A.9}$$

## A.2 Projecting onto the group or algebra

Projections onto the orthogonal/unitary groups or their tangent spaces are central to optimizing over the space of orthogonal/unitary matrices. We focus the discussion here to the case of unitary matrices, but note that all of the following statements apply to orthogonal matrices as well by simple adjustments such as replacing the conjugate transpose ($\dagger$) with the transpose ($\top$). The tangent space $T_{\boldsymbol{U}}$ to a matrix $\boldsymbol{U} \in U(n)$ is equal to

$$T_{\boldsymbol{U}} U(n) = \left\{ \boldsymbol{X} \in \mathbb{C}^{n \times n} : \boldsymbol{U}^\dagger \boldsymbol{X} + \boldsymbol{X}^\dagger \boldsymbol{U} = 0 \right\}. \tag{A.10}$$

Given the canonical inner product $\langle \boldsymbol{A}, \boldsymbol{B} \rangle = \mathrm{Re}\left( \mathrm{Tr}[\boldsymbol{A}^\dagger \boldsymbol{B}] \right)$, we can show that the orthogonal projection onto the tangent space $T_{\boldsymbol{U}} U(n)$ is equal to that given by Lemma 4.3 copied below.

**Lemma 4.3** (Tangent space projection [82]). *Given the tangent space $T_{\boldsymbol{U}} U(n)$ of an orthogonal/unitary matrix $\boldsymbol{U}$, the orthogonal projection $\Pi_{T_{\boldsymbol{U}}}$ with respect to the canonical metric $\langle \boldsymbol{X}, \boldsymbol{Y} \rangle = \mathrm{Re}\left( \mathrm{Tr}[\boldsymbol{X}^\dagger \boldsymbol{Y}] \right)$ is*

$$\Pi_{T_{\boldsymbol{U}}}(\boldsymbol{X}) = \frac{1}{2} \left( \boldsymbol{X} - \boldsymbol{U} \boldsymbol{X}^\dagger \boldsymbol{U} \right). \tag{5}$$

*Similar to Lemma 4.1, this projection also returns the closest matrix in Frobenius norm to $\boldsymbol{X}$ in the tangent space,*

$$\min_{\boldsymbol{Y} \in T_{\boldsymbol{U}} U(n)} \| \boldsymbol{Y} - \boldsymbol{X} \|_F = \Pi_{T_{\boldsymbol{U}}}(\boldsymbol{X}). \tag{6}$$

*Proof.* By the definition of an orthogonal projection, we need that $\Pi_{T_{\boldsymbol{U}}}(\Pi_{T_{\boldsymbol{U}}}(\cdot)) = \Pi_{T_{\boldsymbol{U}}}(\cdot)$ and for all $\boldsymbol{X}, \boldsymbol{Y} \in T_{\boldsymbol{U}} U(n)$ $\Pi_{T_{\boldsymbol{U}}}(\boldsymbol{X}) = \boldsymbol{X}$ and $\langle \Pi_{T_{\boldsymbol{U}}}(\boldsymbol{X}), \boldsymbol{X} - \Pi_{T_{\boldsymbol{U}}}(\boldsymbol{X}) \rangle = 0$. The first two properties are straightforward to check. The last property can be shown as below using the definition of $\Pi_{T_{\boldsymbol{U}}}$ and cyclic property of trace:

$$\begin{aligned}
\langle \Pi_{T_{\boldsymbol{U}}}(\boldsymbol{X}), \boldsymbol{X} - \Pi_{T_{\boldsymbol{U}}}(\boldsymbol{X}) \rangle &= \frac{1}{4} \mathrm{Re}\left( \mathrm{Tr}[(\boldsymbol{X} - \boldsymbol{U} \boldsymbol{X}^\dagger \boldsymbol{U})^\dagger (\boldsymbol{X} + \boldsymbol{U} \boldsymbol{X}^\dagger \boldsymbol{U})] \right) \\
&= \frac{1}{4} \mathrm{Re}\left( \mathrm{Tr}[\boldsymbol{X}^\dagger \boldsymbol{X}] - \mathrm{Tr}[\boldsymbol{U}^\dagger \boldsymbol{X} \boldsymbol{X}^\dagger \boldsymbol{U}] + \mathrm{Tr}[\boldsymbol{X}^\dagger \boldsymbol{U} \boldsymbol{X}^\dagger \boldsymbol{U}] - \mathrm{Tr}[\boldsymbol{U}^\dagger \boldsymbol{X} \boldsymbol{U}^\dagger \boldsymbol{X}] \right) \\
&= \frac{1}{4} \mathrm{Re}\left( \mathrm{Tr}[(\boldsymbol{X}^\dagger \boldsymbol{U})^2] - \mathrm{Tr}[((\boldsymbol{X}^\dagger \boldsymbol{U})^2)^\dagger] \right) \\
&= 0.
\end{aligned} \tag{A.11}$$

Furthermore, we have for all $\boldsymbol{Y} \in T_{\boldsymbol{U}} U(n)$ using triangle inequality and unitary invariance of the Frobenius norm:

$$\begin{aligned}
\| \Pi_{T_{\boldsymbol{U}}}(\boldsymbol{X}) - \boldsymbol{X} \|_F &= \left\| \frac{1}{2}(\boldsymbol{X} - \boldsymbol{U} \boldsymbol{X}^\dagger \boldsymbol{U}) - \boldsymbol{X} \right\|_F \\
&\leq \frac{1}{2} \| \boldsymbol{Y} - \boldsymbol{X} \|_F + \frac{1}{2} \| \boldsymbol{Y} + \boldsymbol{U} \boldsymbol{X}^\dagger \boldsymbol{U} \|_F \\
&= \frac{1}{2} \| \boldsymbol{Y} - \boldsymbol{X} \|_F + \frac{1}{2} \| -\boldsymbol{U} \boldsymbol{Y}^\dagger \boldsymbol{U} + \boldsymbol{U} \boldsymbol{X}^\dagger \boldsymbol{U} \|_F \\
&= \frac{1}{2} \| \boldsymbol{Y} - \boldsymbol{X} \|_F + \frac{1}{2} \| \boldsymbol{Y} - \boldsymbol{X} \|_F \\
&= \| \boldsymbol{Y} - \boldsymbol{X} \|_F,
\end{aligned} \tag{A.12}$$

which proves that $\Pi_{T_{\boldsymbol{U}}}(\cdot)$ projects onto the closest matrix $\boldsymbol{Y} \in T_{\boldsymbol{U}} U(n)$ in Frobenius norm. $\square$

Since the set of unitary/orthogonal matrices does not form a vector space, an orthogonal projection is not a well defined operation in this space. However, it is still valid to ask what is the "closest" unitary/orthogonal matrix to a given matrix in a given norm. This is exactly what is stated in Lemma 4.1 copied from the main text and proven below.

**Lemma 4.1** (Projection onto unitary manifold [46]). *Given a matrix $\boldsymbol{A} \in \mathbb{C}^{n \times n}$:*

$$\Pi_U(\boldsymbol{A}) = \underset{\boldsymbol{U} \in \mathcal{U}(n)}{\arg\min} \| \boldsymbol{A} - \boldsymbol{U} \|_F^2 = \boldsymbol{A}(\boldsymbol{A}^\dagger \boldsymbol{A})^{-\frac{1}{2}}, \tag{3}$$

*where $\mathcal{U}(n)$ indicates the set of $n \times n$ unitary matrices.*

*Proof.* We follow the approach of [46] to prove this result. To shorten our notation, let $\boldsymbol{V} = \boldsymbol{A}(\boldsymbol{A}^\dagger \boldsymbol{A})^{-1/2}$. Given any unitary $\boldsymbol{U} \in U(n)$, let $\boldsymbol{U} = \boldsymbol{M} + \boldsymbol{V}$ for the properly chosen $\boldsymbol{M} \in \mathbb{C}^{n \times n}$. From unitarity of $\boldsymbol{U}$ and $\boldsymbol{V}$, we have

$$\boldsymbol{M}\boldsymbol{V}^\dagger + \boldsymbol{M}\boldsymbol{M}^\dagger + \boldsymbol{V}\boldsymbol{M}^\dagger = 0. \tag{A.13}$$

Then,

$$
\begin{aligned}
\|\boldsymbol{A} - \boldsymbol{U}\|_F^2 &= \|\boldsymbol{A} - \boldsymbol{M} - \boldsymbol{V}\|_F^2 \\
&= \|\boldsymbol{A} - \boldsymbol{V}\|_F^2 + \text{Tr}[\boldsymbol{M}\boldsymbol{V}^\dagger + \boldsymbol{M}\boldsymbol{M}^\dagger + \boldsymbol{V}\boldsymbol{M}^\dagger] - \text{Tr}[\boldsymbol{M}^\dagger \boldsymbol{A} + \boldsymbol{A}^\dagger \boldsymbol{M}] \\
&= \|\boldsymbol{A} - \boldsymbol{V}\|_F^2 - \text{Tr}[\boldsymbol{M}^\dagger \boldsymbol{A} + \boldsymbol{A}^\dagger \boldsymbol{M}] \\
&= \|\boldsymbol{A} - \boldsymbol{V}\|_F^2 - \text{Tr}[\boldsymbol{M}^\dagger \boldsymbol{V}(\boldsymbol{A}^\dagger \boldsymbol{A})^{1/2} + (\boldsymbol{A}^\dagger \boldsymbol{A})^{1/2}\boldsymbol{V}^\dagger \boldsymbol{M}],
\end{aligned} \tag{A.14}
$$

and since from Equation (A.13) we have that $\boldsymbol{M}\boldsymbol{V}^\dagger + \boldsymbol{V}\boldsymbol{M}^\dagger = -\boldsymbol{M}\boldsymbol{M}^\dagger$,

$$\|\boldsymbol{A} - \boldsymbol{U}\|_F^2 = \|\boldsymbol{A} - \boldsymbol{V}\|_F^2 + \text{Tr}[(\boldsymbol{A}^\dagger \boldsymbol{A})^{1/2}\boldsymbol{M}\boldsymbol{M}^\dagger]. \tag{A.15}$$

The second term above is non-negative since $\text{Tr}[(\boldsymbol{A}^\dagger \boldsymbol{A})^{1/2}\boldsymbol{M}\boldsymbol{M}^\dagger] = \text{Tr}[\boldsymbol{M}^\dagger (\boldsymbol{A}^\dagger \boldsymbol{A})^{1/2}\boldsymbol{M}]$ and $(\boldsymbol{A}^\dagger \boldsymbol{A})^{1/2}$ is positive semi-definite. Thus, for all $\boldsymbol{U} \in U(n)$,

$$\|\boldsymbol{A} - \boldsymbol{U}\|_F^2 \geq \|\boldsymbol{A} - \boldsymbol{V}\|_F^2, \tag{A.16}$$

which proves the result. $\square$

# B Review of previous unitary neural network techniques

The integration of orthogonal/unitary matrices into neural networks is broadly aimed at maintaining stability in neural networks with many layers. For vanilla recurrent neural networks, repetitive application of the hidden-to-hidden transformation matrix exponentially amplifies or decays the eigenvalues of the transformation, thus resulting in exponentially large or small gradients. Similarly, in deep network architectures where weight matrices are drawn randomly, the norms of hidden states can similarly grow or decay exponentially with added layers. Enforcing unitarity or orthogonality of neural network layer transformations offers a straightforward method to address these issues of instability since orthogonal/unitary matrices have eigenvalues of unity.

To establish notation and provide motivation for later analysis, consider a RNN whose input is a sequence of vectors $\boldsymbol{x}(t)$ with hidden layer $\boldsymbol{h}(t)$ updated according to the following rule:

$$\boldsymbol{h}^{(t)} = \sigma(\boldsymbol{M}\mathbf{x}^{(t)} + \boldsymbol{W}\boldsymbol{h}^{(t-1)}) \tag{B.1}$$

In the unitary or orthogonal formulation for RNNs, the matrix $\boldsymbol{W}$ in Equation (B.1) is replaced by a unitary or orthogonal matrix $\boldsymbol{U}$. During optimization of a unitary RNN, one must enforce the unitarity or orthogonality of $\boldsymbol{U}$ during training.

Existing methods to parameterize and enforce unitarity/orthogonality in neural network layers can be separated into three categories depending on the method of parameterization. We discuss each of these in detail below:

- **Layer-wise transformations:** among the first methods employed in this line of work, these methods parameterize orthogonal/unitary matrices in a layer-wise fashion where each layer is a parameterized orthogonal/unitary matrix. Example parameterizations include Givens rotations and Householder reflections. These methods are efficient when there are not many layers included in the parameterization and typically not employed when full access to the unitary/orthogonal group is required as full parameterization of an $n \times n$ unitary/orthogonal matrix requires $O(n)$ layers.

- **Lie algebra parameterization:** motivated by the fact that staying on the manifold of the Lie algebra is often easier than staying on the manifold of a Lie group, these methods parameterize the Lie algebra of the unitary/orthogonal groups and later obtain the actual unitary/orthogonal matrix by implementing the matrix exponential map. This matrix exponential map is typically the most costly step in these methods as one can either perform it directly (*e.g.,* using an SVD) or approximate it via Taylor series or Padé approximations which require repetitive application of matrices when applying it to an input. Though we adapt techniques from these methods in our work to efficiently perform gradient updates, we do not parameterize matrices in the Lie algebra.

- **Matrix entry parameterization:** as in typical neural network architectures, this method parameterizes an orthogonal/unitary matrix by directly parameterizing the entries of the matrix. This method is optimal in the "forward" direction since performing the transformation on an input simply requires matrix multiplication (as in vanilla architectures). However, updates to the matrix will no longer maintain unitarity or orthogonality, and one must employ methods to project these updates back onto the unitary/orthogonal manifold. This method is employed in our work.

We now present each of the above methods in the order given.

**Layer-wise transformations**  Early algorithms [44, 5, 66] maintained unitarity by parameterizing a matrix $U$ as a series or layered set of k parameterized unitary transformations:

$$U = F_{uni}^{(1)}(\theta_1)F_{uni}^{(2)}(\theta_2)\cdots F_{uni}^{(k)}(\theta_k). \tag{B.2}$$

where $F_{uni}^{(i)}(\theta_i)$ indicates a transformation that maps parameters $\theta_i$ into a unitary matrix. These transformations include parameterized Givens rotations or Householder reflections. As a concrete example, consider the Givens rotation parameterization which is an orthogonal matrix that performs a rotation of the $i, j$-th dimensions by an amount $\theta$:

$$G(i, j, \theta) = \begin{bmatrix} 1 & \cdots & 0 & \cdots & 0 & \cdots & 0 \\ \vdots & \ddots & \vdots & & \vdots & & \vdots \\ 0 & \cdots & \cos\theta & \cdots & -\sin\theta & \cdots & 0 \\ \vdots & & \vdots & \ddots & \vdots & & \vdots \\ 0 & \cdots & \sin\theta & \cdots & \cos\theta & \cdots & 0 \\ \vdots & & \vdots & & \vdots & \ddots & \vdots \\ 0 & \cdots & 0 & \cdots & 0 & \cdots & 1 \end{bmatrix}, \tag{B.3}$$

or in other words, entries $G_{ii} = G_{jj} = \cos\theta$, $G_{ij} = -G_{ji} = \sin\theta$, and $G_{kl} = \delta_{kl}$ for all other entries.

In general, at least $k = O(n)$ layers are needed to achieve a full parameterization of the $n \times n$ unitary matrices. Training is performed by updating the parameters $\theta_i$ within each layer. However, for large matrices, due to the fact that $O(n)$ layers are required to parameterize the full space of transformations, these algorithms are only efficient when parameterizations over a subset of the space of unitary/orthogonal matrices suffices. Achieving this balance of parameterization versus performance is challenging as prior work – especially work studying the learnability of unitary matrices in quantum computation – has shown that loss landscapes over the unitary/orthogonal manifold may contain many bad local minima [50, 26, 4, 22].

**Lie algebra parameterization**  As a reminder, the Lie algebra of the orthogonal and unitary groups are the set of skew symmetric ($\mathfrak{o}(n)$) and skew Hermitian ($\mathfrak{u}(n)$) matrices,

$$\mathfrak{o}(n) = \left\{ A \in \mathbb{R}^{n \times n} : A + A^\mathsf{T} = 0 \right\}, \tag{B.4}$$

$$\mathfrak{u}(n) = \left\{ A \in \mathbb{C}^{n \times n} : A + A^\dagger = 0 \right\}. \tag{B.5}$$

Transformations from the Lie algebra to the Lie group are performed using the exponential map which is surjective onto the connected components of the identity:

$$\exp(X) = \sum_{k=0}^{\infty} \frac{X^k}{k!} = I + X + \frac{1}{2}X^2 + \frac{1}{6}X^3 + \cdots \tag{B.6}$$

Thus, these methods parameterize the full space of unitary or special orthogonal matrices. However, the orthogonal group has two connected components (matrices with determinant equal to one and negative one) so the exponential maps only onto the positive determinant matrices.

Note that the Lie algebra is a vector space so the sum of two matrices in the Lie algebra is also in the algebra. Thus, gradient updates can typically be very easily performed. However, the exponential map is often expensive to compute, and much prior work has focused on implementing this transformation efficiently. [35] and [73] employ approximations to the matrix exponential via Padé or Taylor series approximations. For example, in the Taylor series approximation, one simply truncates the Taylor series to $K$ entries:

$$\exp(X) \approx \sum_{k=0}^{K} \frac{X^k}{k!}, \quad \left\| \exp(X) - \sum_{k=0}^{K} \frac{X^k}{k!} \right\|_2 \leq \frac{\|X\|_2^k}{k!}, \tag{B.7}$$

where the norm $\|\cdot\|_2$ indicates the spectral norm (*i.e.,* largest singular value). This application of the matrix exponential comes at an added cost both when calculating derivatives and when applying the matrix in the forward direction (*e.g.,* when one simply desires the output of a network). Applying the approximation above to the input of a layer requires $O(K)$ applications of the matrix. The matrix $\exp(\boldsymbol{X})$ can be explicitly constructed to avoid this added cost; however, obtaining this matrix requires a one time cost of $O(K)$ matrix-matrix multiplication operations which can be costly for large matrices.

The Padé approximants are more typically used in approximating the exponential map since they can guarantee that the approximated matrix is actually a unitary/orthogonal matrix (as opposed to a simple Taylor series approximation which does not provide this guarantee). A Padé approximant is an optimal rational function approximation to a given function. The Padé approximant to the exponential map takes the form $\exp(\boldsymbol{A}) \approx r_{mn}(\boldsymbol{A}) = p_m(\boldsymbol{A})q_n(\boldsymbol{A})^{-1}$ where $p_m(\cdot)$ and $q_n(\cdot)$ are order $m$ and $n$ polynomials respectively which have closed forms:

$$p_m(\boldsymbol{A}) = \sum_{k=0}^{m} \frac{(m+n-k)!m!}{(m+n)!(m-k)!k!}\boldsymbol{A}^k, \quad q_n(\boldsymbol{A}) = \sum_{k=0}^{n} \frac{(m+n-k)!n!}{(m+n)!(n-k)!k!}(-\boldsymbol{A})^k. \tag{B.8}$$

The error of the approximation scales as $O(\|\boldsymbol{A}\|^{m+n+1})$ [37]. For unitary/orthogonal matrices, this approximation has the feature that for order $m = n$, applying the approximation to an element of the Lie algebra of the orthogonal/unitary matrices outputs a orthogonal/unitary matrix. Notably, setting $m = n = 1$ obtains the Cayley transform which was used in [35].

Finally, we note that approximations to the exponential function often can bias the gradients so in the RNN setting, [56] actually perform an SVD to obtain the actual output of the exponential map and analytically calculate gradients.

**Matrix entry parameterization** Parameterizing the entries of a matrix $\boldsymbol{U}$ directly is an obvious and simple means of constructing a unitary or orthogonal matrix. However, the set of unitary/orthogonal matrices do not form an algebra so one cannot simply update these matrices by simply adding a gradient update to a given matrix. Instead, updates to these matrices must be performed by projecting the updated matrix back onto the set of unitary/orthogonal matrices. Obviously, $\boldsymbol{U}$ must be initialized to be unitary/orthogonal in these methods.

Given a gradient update $\boldsymbol{G}$, the matrix $\boldsymbol{U} + \boldsymbol{G}$ is typically no longer unitary. Methods of Riemannian optimization are employed to update the matrix in a unitary/orthogonal fashion. One means of updating the matrix is via the Cayley transform which computes a parametric curve in the direction, employed in [57, 82, 61, 76]. The Cayley transform "transports" $\boldsymbol{U}$ in the direction of the projection of the gradient in the tangent space $\Pi(\boldsymbol{G})$. *i.e.,* let $\boldsymbol{U}(0)$ be the initial unitary matrix, then one can transport the matrix in the direction of $\Pi(\boldsymbol{G})$ (where $\Pi(\cdot)$ is the projection onto the tangent space) by a "length" of $\alpha$ via the Cayley transform [67],

$$\boldsymbol{U}(\alpha) = \left(\boldsymbol{I} - \frac{\alpha}{2}\Pi(\boldsymbol{G})\right)^{-1}\left(\boldsymbol{I} + \frac{\alpha}{2}\Pi(\boldsymbol{G})\right)\boldsymbol{U}(0). \tag{B.9}$$

The above update formula requires a matrix inversion step which is the most costly step for large matrices. [57, 61] approximate the transformation via a fixed point interation which avoids having to invert a matrix but still requires matrix-matrix multiplication. More generally, the Cayley transform is a first order Padé approximant to the exponential map which provides a connection between the matrix entry parameterization and the Lie algebra parameterization discussed previously [37], *i.e.,* setting $m = n = 1$ in Equation (B.8) obtains the Cayley transform.

Our algorithms described in the main text directly parameterize matrix entries and thus follow this parameterization method. In performing the exponential map, we do not resort to any approximations since the exponential is efficient to perform in low rank settings. Updates to unitary matrices are either projected directly onto the closest unitary matrix in Frobenius norm (PROJUNN-D) or transported along the geodesic in the direction of the projection of the gradient onto the tangent space (PROJUNN-T). We refer the reader to the main text for a full description of our methodology.

## B.1 Orthogonal or unitary convolution

Since convolutions are linear operators, one can perform orthogonal or unitary convolutions using similar techniques as in the general case studied above. As a reminder, for 2-D convolution, given input tensor $\mathbf{X} \in \mathbb{C}^{M \times N \times C}$ where $C$ denotes the number of channels of the $M \times N$ input, linear convolution (or technically cross-correlation) with a filter $\mathbf{W} \in \mathbb{C}^{M \times N \times C \times C}$ takes the form

$$[\text{conv}_{\mathbf{W}}(\mathbf{X})]_{p,q,d} = [\mathbf{W} * \mathbf{X}]_{p,q,d} = \sum_{c=1}^{C}\sum_{m=1}^{M}\sum_{n=1}^{N} \mathbf{W}_{m,n,d,c}\mathbf{X}_{p+m,q+n,c}, \tag{B.10}$$

where the indexing above is assumed to be cyclic (taken modulus the corresponding dimension). Unitary or orthogonal convolutions form a subset of filters $\mathbf{W}$ which preserve the norm, *i.e.,* $\| \text{conv}_\mathbf{W}(\mathbf{X})\| = \|\mathbf{X}\|$. Equivalently, $\text{conv}_\mathbf{W}(\cdot)$ is orthogonal/unitary if the Jacobian of the transformation is also orthogonal/unitary.

The first method for orthogonalizing convolutions in neural networks was proposed in [72]. Their algorithm showed how to calculate the singular values of a linear convolution operation and then performed a series of projections on a given convolutional filter to project it onto an operator norm ball. Their algorithm made use of the convolution theorem which shows that linear convolution is diagonalized by Fourier transformations. [59] also proposed a method to orthogonalize convolutions by performing an SVD of the linearization of a convolution and then bounding the singular values accordingly. This method outputs linear transformations that are close to a convolution operation but no longer necessarily maintaining the equivariance of a standard convolution. Also, their method is expensive for large matrices as it requires implenting an SVD.

More recent methods implement orthogonal convolutions via approximations to the exponential map [76, 73]. These methods first form convolution filters $\mathbf{W}$ whose Jacobians $\mathbf{J}(\mathbf{W})$ satisfy the skew symmetry property of the orthogonal group:

$$\mathbf{J}(\mathbf{W}) = -\mathbf{J}(\mathbf{W})^\mathsf{T} \equiv \mathbf{W} = -\text{conv-transpose}(\mathbf{W}), \tag{B.11}$$

where conv-transpose is the equivalent transposition operation in the filter space defined as [73]

$$[\text{conv-transpose}(\mathbf{L})]_{m,n,c,d} = \mathbf{L}^*_{M-1-m,N-1-n,d,c} \tag{B.12}$$

for a filter $\mathbf{L} \in \mathbb{R}^{M \times N \times D \times C}$.

A given filter $\mathbf{W}$ can be transformed into a skew symmetric convolution filter by simply applying $\mathbf{L} = \mathbf{W} - \text{conv-transpose}(\mathbf{W})$. As in the general case, one can apply the exponential map to the convolution filter to perform orthogonal convolution.

$$\exp[\text{conv}_\mathbf{L}](\mathbf{X}) = \mathbf{X} + \mathbf{L} * \mathbf{X} + \frac{1}{2}\mathbf{L} *^2 \mathbf{X} + \frac{1}{6}\mathbf{L} *^3 \mathbf{X} + \cdots , \tag{B.13}$$

where $*^p$ indicates the convolution is applied $p$ times, *e.g.,* $\mathbf{L} *^2 \mathbf{X} = \mathbf{L} * \mathbf{L} * \mathbf{X}$. Since the above operation is computationally expensive [73] implement a $k$-th order Taylor approximation to the exponential map:

$$\exp[\text{conv}_\mathbf{L}](\mathbf{X}) \approx \sum_{p=0}^{k} \frac{1}{p!}\mathbf{L} *^p \mathbf{X}. \tag{B.14}$$

Given 2-D inputs of dimension $N \times N \times C$, applying convolution with filters with support over $W$ elements in each dimension has runtime $O(pC^2W^2N^2)$. The added factor $p$ in the runtime is required both upon training and evaluation of the network. Though this number is held constant, this adds a multiplicative overhead to running the algorithm both when evaluating an input and when training the algorithm. [73] use the above method to implement orthogonal convolution in their skew orthogonal convolution (SOC) algorithm. Similar techniques have been used to perform invertible (not necessarily orthogonal or unitary) convolutions in [39].

There are two key considerations or drawbacks associated with using the Taylor expansion to perform unitary/orthogonal convolution as in [73]. First, approximations via the Taylor approximation necessarily include an error that can be accounted for by scaling the factor $p$. However, unlike the Padé approximants discussed earlier such as the Cayley approximant, the Taylor approximation to the exponential map does not return a unitary/orthogonal operator. Thus, the value $p$ needed to bound the error from any unitary/orthogonal operator also scales logarithmically with the desired error; however, this factor can also grow with the size of the filter, the number of channels, size of the input to a convolution operation, and especially the depth of the network. Second, Taylor approximations to a function can significantly bias the gradient. As noted in [56], their expRNN algorithm avoided using the Taylor or Padé approximation in the implementation of the unitary/orthogonal operation for this reason. As an example of how approximations can fail to represent the gradient, they provide the set of functions $f_n(x) = \sin(2\pi nx)/n$ approximating $f = 0$, where $f_n \to f$ uniformly but the derivatives do not converge to zero. Especially when constructing very deep networks, such instabilities can potentially cause issues in training.

Convolution operations over filters with a large support can be performed more efficiently in the Fourier domain as explored in various prior work [64, 11]. Specific to orthogonal convolution, [76] use the 2-D fast Fourier transform to more efficiently perform orthogonal convolution. For single channel inputs and outputs, convolution in the Fourier regime corresponds to pointwise multiplication over Fourier bases. Given multi-channel inputs and outputs, convolution in the Fourier regime corresponds to matrix multiplication over the blocks indexed by channels where each block corresponds to a specific Fourier basis, *i.e.,* for a filter $\mathbf{W}$ and input $\mathbf{X}$:

$$[\text{FFT} \, \text{conv}_\mathbf{W}(\mathbf{X})]_{\widehat{r},\widehat{s},:} = \widehat{\mathbf{W}}^*_{\widehat{r},\widehat{s},:,:} \, [\text{FFT} \, \mathbf{X}]_{\widehat{r},\widehat{s},:} \, , \tag{B.15}$$

where $\widehat{\mathbf{W}}_{\widehat{r},\widehat{s},:,:}$ is the representation of the filter in the Fourier regime. If the filter and and input are flattened in their spatial dimensions, we can represent the operation above as block-diagonal multiplication. For example, one can visually input the Fourier multiplication on an 2-channel input with spatial dimension $[2, 2]$.

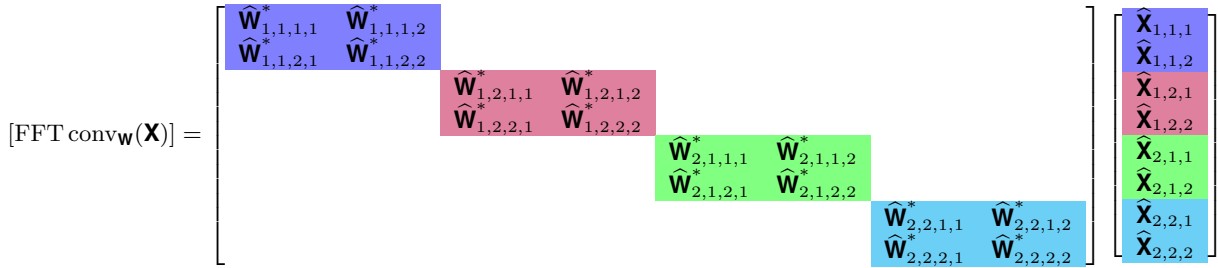

[76] use the above technique, parameterizing filters in the Lie algebra of the orthogonal group. Filters are mapped into the orthogonal group using the Cayley transform (see Equation (B.9) for its form). Since the Cayley transform requires matrix inversion over the matrices above, their total runtime scales as $O(N^2C^2 \log(N) + N^2C^3)$ operations for convolution over $N \times N$ images with $C$ channels. Our methodology is adapted from the techniques in [76] and scales more efficiently as $O(N^2C \log(N) + kN^2C^2)$ time when the rank of gradient updates $k \ll N$. We instead parameterize the convolution filter directly in the Fourier regime, by parameterizing the individual blocks ($\widehat{\mathbf{W}}_{\widehat{r},\widehat{s},:,:}$) of the block diagonal matrix above. One drawback of our method compared to [76] is that we cannot specify the support of the convolution in the real space of the Lie algebra as we parameterize matrices directly in the Fourier space. Of course, one can perform projections of the convolutions in Fourier space onto the bases spanned by the support of the elements in the Lie algebra; however, we opt instead to implement a regularizer which biases the filter towards elements of the desired space (*e.g.,* local elements).

Finally, [58] performed orthogonal convolution by parameterizing a block of matrices and corresponding projectors in their Block Convolutional Orthogonal Parametrization (BCOP) algorithm. Their method preceded those of [76] and [73]. However, their method has two key drawbacks: it only parameterizes a subset of the space of orthogonal convolutions and is slower than other methods as it adds additional parameters to connect the various components in the space of orthogonal convolution.

Finally, methods have been proposed to iteratively or approximately maintain orthogonality. [9] propose a regularizer for convolutional layers which penalize matrices having singular values far from unity. [40] propose using a Newton's method iteration to update linear transformations to be closer to orthogonal/unitary. Their method requires iterative matrix-matrix multiplication over the unrolled weight matrix which can become expensive for large images.

## B.2   Other related works

In the context of recurrent neural networks, designing RNNs to learn long sequences of data has a rich history of study. Some of the first and most celebrated algorithms include the long short-term memory networks (LSTM) [38] and gated reccurent unit (GRU) networks [20]. Since these works, various techniques have been used to more optimally avoid issues with learning long-sequence data. Beyond the unitary RNNs discussed earlier, some work has explored using bi-directional RNNs [33, 53], including those that have a more biologically inspired design [12]. These networks perform well on the copy task.

Another line of research studies the stability properties of continuous state space models which can be converted into a RNN formulation [24]. Some algorithms construct continuous state space models whose attractors are stable points in the dynamical system [19, 25]. More recently, continuous state space models have been designed with hidden state transformations that are customized to memorize data by limiting the learning over time to a subset of orthogonal polynomials [29, 30, 77]. Here, hidden states are in a sense parameterized over a set of polynomial coefficients [77]. These algorithms perform very well on tasks such as the copy task or Permuted MNIST but do not include unitary/orthogonal transformations in their network. In fact, more recent models [30, 77] achieve slightly higher scores on the TIMIT and permuted MNIST benchmarks compared to the unitary RNN formulations studied here.

In the convolutional setting, [83] form very deep convolutional neural networks by initializing parameters to construct norm-preserving orthogonal transformations. Orthogonality is not preserved during training however. [80] bias filters towards orthogonality by implementing a regularizer that penalizes the weights when norms of outputs are larger or smaller than norms of inputs. The networks used in [83, 80] do not necessarily preserve the orthogonality property of their convolutional transformations during training. [45] study orthogonal convolutions from the basis of unitary/orthogonal wavelets showing how to represent convolution in terms of these wavelets.

Low rank approximations have been used in prior work in deep learning to prune neural network models [84, 74] and accelerate convolutions [43, 75, 42]. More related to our work, recent research has compressed gradients efficiently using low rank compression methods [78]. Although their focus was in sharing compressed information across computing units, their work lends support to the notion that the information of a gradient can be effectively and efficiently stored in low rank components. Research in learning theory has also noted connections between the stable rank of neural network parameters and generalization. [7] prove a generalization bound by compressing the models in the hypothesis class based on the stable rank of individual layers. [63] study the phases of learning from a random matrix theory setting showing that the stable rank of a matrix tends to decay over training. Various works have studied the implicit bias induced by optimization algorithms such as gradient descent showing that in many cases, the implicit bias is towards low rank solutions [31, 21]. Such bias towards low rank solutions has been explicitly proven in the setting of 2-layer matrix factorization [60], deep matrix factorization [6], linear group convolutional networks [54], and matrix recovery from Pauli measurements [62]. We note that relating low-rankness to the generalization ability of learning algorithms is a richly studied topic, and there are numerous papers that we did not mention here.

Finally, a wide range of work in quantum computation and quantum machine learning studies unitary learning algorithms in the context of quantum systems [13]. In fact, since the state space of closed quantum systems is transformed by unitary operators, quantum computers offer a unique platform for performing machine learning on the unitary manifold. This is an active area of research and existing methods for performing quantum machine learning on quantum architectures include variational algorithms which parameterize a quantum circuit and update the parameters via classical optimization methods [17, 81, 85, 48, 49, 79], quantum neural networks which design analogues to classical deep neural networks [51, 10, 71], and direct implementations of classical deep learning algorithms on quantum architectures [47, 16, 3]. We stress that these quantum algorithms are inherently different in nature than their classical counterparts and there still exist significant challenges that must be surmounted before they become practically feasible [17, 13, 65]. For example, the loss landscapes of quantum algorithms often have many more poor local minima in comparison to classical counterparts [50, 4] and training of quantum architectures requires sampling of outputs which is challenging when derivatives decay with the size of a model – a phenomenon described as "barren plateaus" [65, 18]. Furthermore, even if algorithms can be efficiently run on a quantum computer, preparing data for use in a quantum computer and reading out information from the quantum computer are challenging tasks which are not guaranteed to be efficient [1].

## C   Deferred proofs

### C.1   Proof of Theorem 4.2

Recall Theorem 4.2:

**Theorem 4.2** (Low rank unitary projection). *Let $U$ be an $n \times n$ orthogonal/unitary matrix perturbed by $G_k$, a rank $k$ matrix. Then the projection onto the closest orthogonal/unitary matrix defined below can be performed in $O(k(n^2 + nk + k^2))$ steps.*

$$U + G_k \to \underset{V \in \mathcal{U}}{\arg\min} \|U + G_k - V\|_F^2. \tag{4}$$

*Proof.* We proceed to prove the above statement by first analyzing the case where $k = 1$ and then generalizing to higher rank $k$. Recall from Lemma 4.1 that we would like to perform the following update:

$$U + G_k \to \underset{V \in \mathcal{U}}{\arg\min} \|(U + G_k) - V\|_F^2 = (U + G_k) \left[ (U + G_k)^\dagger (U + G_k) \right]^{-\frac{1}{2}}. \tag{C.1}$$

Let the rank one vector components of $G_k = ab^\dagger$ and define

$$\tilde{M} = U + G_k = U + ab^\dagger. \tag{C.2}$$

With the above, we can rewrite $(\tilde{M}^\dagger \tilde{M})^{-\frac{1}{2}}$ in Equation (C.1) as:

$$\begin{aligned} (\tilde{M}^\dagger \tilde{M})^{-\frac{1}{2}} &= [(U + ab^\dagger)^\dagger (U + ab^\dagger)]^{-\frac{1}{2}} \\ &= [I + \hat{a}b^\dagger + b\hat{a}^\dagger + c_a bb^\dagger]^{-\frac{1}{2}}, \end{aligned} \tag{C.3}$$

where $\hat{a} = \tilde{U}^\dagger a$ and $c_a = a^\dagger a$.

Equation (C.3) is the Identity matrix plus the update of a rank two matrix. To see this, we decompose $\hat{a}$ and $b$ into orthogonal components using Gram Schmidt:

$$v_1 = \frac{b}{\|b\|} \quad v_2 = \frac{\hat{a} - (v_1^\dagger \hat{a})\hat{a}}{\|\hat{a} - (v_1^\dagger \hat{a})\hat{a}\|}. \tag{C.4}$$

In this new basis:

$$\hat{a} = a_1 v_1 + a_2 v_2 \quad b = b_1 v_1, \tag{C.5}$$

and

$$\tilde{M}^\dagger \tilde{M} =$$
$$I + \begin{bmatrix} v_1 & v_2 \end{bmatrix} \begin{bmatrix} a_1 b_1^* + b_1 a_1^* + c_a b_1 b_1^* & b_1 a_2^* \\ a_2 b_1^* & 0 \end{bmatrix} \begin{bmatrix} v_1^\dagger \\ v_2^\dagger \end{bmatrix}. \tag{C.6}$$

Performing an eigendecomposition of the above $2 \times 2$ matrix into a diagonal eigenvalue matrix $S$ and eigenvector matrix $C$, we can rewrite $\tilde{M}^\dagger \tilde{M}$ in a convenient form:

$$\tilde{M}^\dagger \tilde{M} = I + \begin{bmatrix} v_1 & v_2 \end{bmatrix} C \begin{bmatrix} s_1 & 0 \\ 0 & s_2 \end{bmatrix} C^\dagger \begin{bmatrix} v_1^\dagger \\ v_2^\dagger \end{bmatrix}$$
$$= I + s_1 u_1 u_1^\dagger + s_2 u_2 u_2^\dagger. \tag{C.7}$$

Taking the inverse square root of the above can be performed by manipulating singular values:

$$(\tilde{M}^\dagger \tilde{M})^{-\frac{1}{2}} = I + ((s_1 + 1)^{-\frac{1}{2}} - 1)u_1 u_1^\dagger$$
$$+ ((s_2 + 1)^{-\frac{1}{2}} - 1)u_2 u_2^\dagger. \tag{C.8}$$

Finally, we multiply the above on the left by $\tilde{M}$:

$$\tilde{M}(\tilde{M}^\dagger \tilde{M})^{-\frac{1}{2}} = \tilde{M} + ((s_1 + 1)^{-\frac{1}{2}} - 1)\tilde{M} u_1 u_1^\dagger$$
$$+ ((s_2 + 1)^{-\frac{1}{2}} - 1)\tilde{M} u_2 u_2^\dagger. \tag{C.9}$$

The above requires performing an eigendecomposition of a $2 \times 2$ matrix and a series of matrix-vector multiplication, matrix additions, and vector-vector outer products – in total scaling as $O(n^2)$ time.

**Rank $k$ updates** Note that the above method can be extended to low rank updates, running in $O(kn^2)$ time when the rank of the update $k \ll n$. Specifically, now our update is:

$$\tilde{M} = U + \sum_{i=1}^{k} a_i b_i^\dagger. \tag{C.10}$$

Following the same steps would ultimately require performing an eigendecomposition of a $2k \times 2k$ matrix which takes $O(k^3)$ time. Additionally, one must perform Gram-Schmidt decomposition on a set of $k$ vectors of length $n$ which takes $O(k^2 n)$ time. Finally, a series of $O(k)$ matrix-vector multiplications and vector-vector outer products is performed resulting in a total runtime of $O(k(n^2 + nk + k^2))$ time. In cases where $k \ll n$, the time to perform the Gram-Schmidt decomposition and the eigendecomposition is negligible and an overall runtime of $O(kn^2)$ time is achieved. Even in cases where updates are not low rank, one can apply efficient sampling procedures (see Section 4.3) to find low rank approximations to the update matrix and apply the methods above while maintaining runtimes.

$\square$

## C.2 Proof of Theorem 4.4

Recall Theorem 4.4:

**Theorem 4.4** (Low rank tangent transport)**.** *Let $U$ be an $n \times n$ orthogonal/unitary matrix perturbed by $G_k$, a rank $k$ matrix. Then projecting $G_k$ onto the tangent space and performing a rotation in that direction as defined in Equation (7) can be performed in $O(k(n^2 + nk + k^2))$ steps.*

*Proof.* We would like to efficiently perform the update below:

$$U \to U \exp \left[ -\eta U^\dagger \Pi_{T_U}(G_k) \right]. \tag{C.11}$$

As before, we proceed to prove the above statement by first analyzing the case where $k = 1$ and then generalizing to higher rank $k$.

Let the rank one vector components of $G_k = ab^\dagger$. Then

$$U^\dagger \Pi_{T_U}(G_k) = \frac{1}{2} U^\dagger \left( G_k - U G_k^\dagger U \right) = \frac{1}{2} \left( U^\dagger ab^\dagger - ba^\dagger U \right) = \frac{1}{2} \left( \hat{a}b^\dagger - b\hat{a}^\dagger \right), \tag{C.12}$$

where $\hat{a} = U^\dagger a$. The above is a rank 2 matrix. As before, we now proceed to perform an eigendecomposition in the low rank subspace of the above matrix. Using Gram Schmidt, we have

$$v_1 = \frac{b}{\|b\|} \quad v_2 = \frac{\hat{a} - (v_1^\dagger \hat{a})\hat{a}}{\|\hat{a} - (v_1^\dagger \hat{a})\hat{a}\|}. \tag{C.13}$$

In this new basis:

$$\hat{a} = a_1 v_1 + a_2 v_2 \quad b = b_1 v_1, \tag{C.14}$$

and

$$U^\dagger \Pi_{T_U}(G_k) = \frac{1}{2} \begin{bmatrix} v_1 & v_2 \end{bmatrix} \begin{bmatrix} a_1 b_1^* - b_1 a_1^* & -b_1 a_2^* \\ a_2 b_1^* & 0 \end{bmatrix} \begin{bmatrix} v_1^\dagger \\ v_2^\dagger \end{bmatrix}. \tag{C.15}$$

Performing an eigendecomposition of the above $2 \times 2$ matrix into a diagonal eigenvalue matrix $S$ and eigenvector matrix $C$, we can rewrite $\tilde{M}^\dagger \tilde{M}$ in a convenient form:

$$\begin{aligned} U^\dagger \Pi_{T_U}(G_k) &= \begin{bmatrix} v_1 & v_2 \end{bmatrix} C \begin{bmatrix} s_1 & 0 \\ 0 & s_2 \end{bmatrix} C^\dagger \begin{bmatrix} v_1^\dagger \\ v_2^\dagger \end{bmatrix} \\ &= s_1 u_1 u_1^\dagger + s_2 u_2 u_2^\dagger. \end{aligned} \tag{C.16}$$

We apply the exponential map scaled by $\eta$ to obtain

$$\begin{aligned} \exp \left[ -\eta U^\dagger \Pi_{T_U}(G_k) \right] = I &+ (\exp(-\eta s_1) - 1) u_1 u_1^\dagger \\ &+ (\exp(-\eta s_2) - 1) u_2 u_2^\dagger. \end{aligned} \tag{C.17}$$

Finally, we multiply the above on the left by $U$ to obtain the final result

$$U \exp \left[ -\eta U^\dagger \Pi_{T_U}(G_k) \right] = U + (\exp(-\eta s_1) - 1) U u_1 u_1^\dagger + (\exp(-\eta s_2) - 1) U u_2 u_2^\dagger. \tag{C.18}$$

The above requires performing an eigendecomposition of a $2 \times 2$ matrix and a series of matrix-vector multiplication, matrix additions, and vector-vector outer products – in total scaling as $O(n^2)$ time.

**Rank $k$ updates**  Note that as before, the above method can be extended to low rank updates, running in $O(kn^2)$ time when the rank of the update $k \ll n$. Specifically, now our update is:

$$U \to U \exp \left[ -\eta U^\dagger \Pi_{T_U} \left( \sum_{i=1}^{k} a_i b_i^\dagger \right) \right]. \tag{C.19}$$

Following the same steps, the projection onto the tangent space would be a rank $2k$ matrix. The steps that follow would ultimately require performing an eigendecomposition of a $2k \times 2k$ matrix which takes $O(k^3)$ time. Additionally, one

must perform Gram-Schmidt decomposition on a set of $k$ vectors of length $n$ which takes $O(k^2 n)$ time. Finally, a series of $O(k)$ matrix-vector multiplications and vector-vector outer products is performed resulting in a total runtime of $O(k(n^2 + nk + k^2))$ time. In cases where $k \ll n$, the time to perform the Gram-Schmidt decomposition and the eigendecomposition is negligible and an overall runtime of $O(kn^2)$ time is achieved. Even in cases where updates are not low rank, one can apply efficient sampling procedures (see Section 4.3) to find low rank approximations to the update matrix and apply the methods above while maintaining runtimes.

$\square$

### C.3   First order equivalence to PROJUNN-D

Though PROJUNN-D and PROJUNN-T perform different updates, we can show that up to first order, the updates are in fact equivalent. Furthermore, as one may expect, this first order update is equal to the projection of the gradient onto the tangent space (see Lemma 4.3). In the case of PROJUNN-D, this shows that the update step is a retraction or first order approximation to the matrix exponential implemented in PROJUNN-T [15].

**Proposition C.1** (First order equivalence). *For an $n \times n$ unitary/orthogonal matrix $\boldsymbol{U}$ perturbed by $\Delta \boldsymbol{U}$, gradient updates applied by algorithms* PROJUNN-D *(Equation* (4)*) and* PROJUNN-T *(Equation* (7)*) are, up to first order, equal to $\boldsymbol{U} + \Pi_{T_U}(\Delta \boldsymbol{U})$, i.e.,*

$$\boldsymbol{U} \to \boldsymbol{U} + \frac{1}{2}\left(\Delta \boldsymbol{U} - \boldsymbol{U}\Delta \boldsymbol{U}^\dagger \boldsymbol{U}\right) + O(\Delta \boldsymbol{U}\Delta \boldsymbol{U}^\dagger).$$

*Proof.* We first show that the above formula holds for PROJUNN-D and then show the same for PROJUNN-T. Recall the update formula for PROJUNN-D:

$$\boldsymbol{U} + \Delta \boldsymbol{U} \to \underset{\boldsymbol{V} \in \mathcal{U}}{\arg\min} \|(\boldsymbol{U} + \Delta \boldsymbol{U}) - \boldsymbol{V}\|_F^2 = (\boldsymbol{U} + \Delta \boldsymbol{U})\left[(\boldsymbol{U} + \Delta \boldsymbol{U})^\dagger (\boldsymbol{U} + \Delta \boldsymbol{U})\right]^{-\frac{1}{2}}. \qquad \text{(C.20)}$$

Expanding the above up to first order and applying the first order Taylor expansion of $(\cdot)^{-1/2}$, we have:

$$
\begin{aligned}
(\boldsymbol{U} + \Delta \boldsymbol{U})\left[(\boldsymbol{U} + \Delta \boldsymbol{U})^\dagger (\boldsymbol{U} + \Delta \boldsymbol{U})\right]^{-\frac{1}{2}} &= (\boldsymbol{U} + \Delta \boldsymbol{U})\left[\boldsymbol{I} + \boldsymbol{U}^\dagger \Delta \boldsymbol{U} + \Delta \boldsymbol{U}^\dagger \boldsymbol{U} + O(\Delta \boldsymbol{U}^\dagger \Delta \boldsymbol{U})\right]^{-\frac{1}{2}} \\
&= (\boldsymbol{U} + \Delta \boldsymbol{U})\left[\boldsymbol{I} - \frac{1}{2}\left(\boldsymbol{U}^\dagger \Delta \boldsymbol{U} + \Delta \boldsymbol{U}^\dagger \boldsymbol{U}\right) + O(\Delta \boldsymbol{U}^\dagger \Delta \boldsymbol{U})\right] \\
&= \boldsymbol{U} + \frac{1}{2}\left(\Delta \boldsymbol{U} - \boldsymbol{U}\Delta \boldsymbol{U}^\dagger \boldsymbol{U}\right) + O(\Delta \boldsymbol{U}^\dagger \Delta \boldsymbol{U}).
\end{aligned}
\qquad \text{(C.21)}
$$

Similarly, for PROJUNN-T, recall the update formula (ignoring $-\eta$ term for learning rate):

$$\boldsymbol{U} \to \boldsymbol{U} \exp\left[\boldsymbol{U}^\dagger \Pi_{T_U}(\Delta \boldsymbol{U})\right]. \qquad \text{(C.22)}$$

Expanding the above and applying the first order approximation of the matrix exponential,

$$
\begin{aligned}
\boldsymbol{U} \exp\left[\boldsymbol{U}^\dagger \Pi_{T_U}(\Delta \boldsymbol{U})\right] &= \boldsymbol{U} \exp\left[-\eta \boldsymbol{U}^\dagger \frac{1}{2}\left(\Delta \boldsymbol{U} - \boldsymbol{U}\Delta \boldsymbol{U}^\dagger \boldsymbol{U}\right)\right] \\
&= \boldsymbol{U}\left[\boldsymbol{I} + \frac{1}{2}\left(\Delta \boldsymbol{U} - \boldsymbol{U}\Delta \boldsymbol{U}^\dagger \boldsymbol{U}\right) + O(\Delta \boldsymbol{U}^\dagger \Delta \boldsymbol{U})\right] \\
&= \boldsymbol{U} + \frac{1}{2}\left(\Delta \boldsymbol{U} - \boldsymbol{U}\Delta \boldsymbol{U}^\dagger \boldsymbol{U}\right) + O(\Delta \boldsymbol{U}^\dagger \Delta \boldsymbol{U}).
\end{aligned}
\qquad \text{(C.23)}
$$

$\square$

### C.4   Unitary convolutional manifold is connected

Before proceeding to prove that the space of unitary convolutions is connected, we first provide a version of the convolution theorem which shows that convolution in the Fourier domain corresponds to block multiplication over channels. This is a classic result also contained in various prior works [72, 76], and we provide a short proof here for completeness.

**Lemma C.2.** *Given convolution filter* $\mathbf{W} \in \mathbb{C}^{M \times N \times C \times C}$ *and input* $\mathbf{X} \in \mathbb{C}^{M \times N \times C}$, *recall the definition of cyclic convolution (or technically cross-correlation) of* $\mathbf{W}$ *and* $\mathbf{X}$:

$$[\mathrm{conv}_{\mathbf{W}}(\mathbf{X})]_{p,q,d} = \sum_{c=1}^{C} \sum_{m=1}^{M} \sum_{n=1}^{N} \mathbf{W}_{m,n,d,c} \mathbf{X}_{p+m,q+n,c}. \tag{C.24}$$

*Let* FFT *be the two dimensional fast Fourier transform, then convolution in the Fourier domain corresponds to block-wise multiplication over channels:*

$$[\mathrm{FFT}\,\mathrm{conv}_{\mathbf{W}}(\mathbf{X})]_{\widehat{r},\widehat{s},:} = \widehat{\mathbf{W}}^{*}_{\widehat{r},\widehat{s},:,:} \; [\mathrm{FFT}\,\mathbf{X}]_{\widehat{r},\widehat{s},:}. \tag{C.25}$$

*Proof.* Let the roots of unity be denoted as $\omega_M = e^{2\pi i/M}$ and $\omega_N = e^{2\pi i/N}$. Then,

$$
\begin{aligned}
[\mathrm{FFT}\,\mathrm{conv}_{\mathbf{W}}(\mathbf{X})]_{\widehat{r},\widehat{s},d} &= \sum_{u=1}^{M} \sum_{v=1}^{N} \omega_M^{u\widehat{r}} \omega_M^{v\widehat{s}} \sum_{m=1}^{M} \sum_{n=1}^{N} \sum_{c=1}^{C} \mathbf{W}_{m,n,d,c} \mathbf{X}_{u+m,v+n,c} \\
&= \sum_{u=1}^{M} \sum_{v=1}^{N} \sum_{m=1}^{M} \sum_{n=1}^{N} \sum_{c=1}^{C} \omega_M^{u\widehat{r}} \omega_M^{v\widehat{s}} \omega_M^{\widehat{r}m} \omega_M^{-\widehat{r}m} \omega_N^{\widehat{s}n} \omega_N^{-\widehat{s}n} \mathbf{W}_{m,n,d,c} \mathbf{X}_{u+m,v+n,c} \\
&= \sum_{c=1}^{C} \sum_{m=1}^{M} \sum_{n=1}^{N} \omega_M^{-\widehat{r}m} \omega_N^{-\widehat{s}n} \mathbf{W}_{m,n,d,c} \sum_{u=1}^{M} \sum_{v=1}^{N} \omega_M^{\widehat{r}(u+m)} \omega_M^{\widehat{s}(v+n)} \mathbf{X}_{u+m,v+n,c} \mathbf{X}_{u+m,v+n,c} \\
&= \sum_{c=1}^{C} \widehat{\mathbf{W}}^{*}_{\widehat{r},\widehat{s},d,c} \widehat{\mathbf{X}}_{\widehat{r},\widehat{s},c}.
\end{aligned}
\tag{C.26}
$$

$\square$

As an aside, the complex conjugation of the filter above is due to the fact that convolution in neural networks corresponds to the more commonly used term cross-correlation in mathematics. If the convolution operation were redefined as what is more commonly known as convolution in mathematics, *i.e.,* define $\mathrm{conv}'$ as $[\mathrm{conv}'_{\mathbf{W}}(\mathbf{X})]_{p,q,d} = \sum_{c=1}^{C} \sum_{m=1}^{M} \sum_{n=1}^{N} \mathbf{W}_{m,n,d,c} \mathbf{X}_{p-m,q-n,c}$, then the complex conjugate would no longer appear on the filter term.

From here, we simply show that each block in the above is connected which allows us to prove that the unitary manifold is connected.

**Theorem 4.5** (Unitary convolutional manifold is connected). *The space of unitary convolutions with filters of full support has a single connected component.*

*Proof.* Since the fourier transform is unitary and invertible, we can represent every filter $\mathbf{W}$ in the fourier domain as $\widehat{\mathbf{W}} \in \mathbb{C}^{M \times N \times C \times C}$ and vice-versa. In the Fourier domain, given Lemma C.2, we have that convolution corresponds to block-wise multiplication over channels. For unitary convolution, each block $\widehat{W}_{\widehat{r},\widehat{s},:,:}$ indexed by frequencies $\widehat{r}$ and $\widehat{s}$ must be a unitary matrix, so we now analyze the set of filters whose blocks are unitary matrices in the Fourier domain.

The space of unitary matrices $U(C)$ is connected [32]. Therefore for every block, there exists a connected path between any two unitary matrices $\widehat{W}^{(1)}_{\widehat{r},\widehat{s},:,:}, \widehat{W}^{(2)}_{\widehat{r},\widehat{s},:,:} \in U(C)$. The full space of unitary convolutions is parameterized by the $MN$ times direct product of groups $U(C)$, *i.e.,* $U(C)^{(\times MN)}$. Since the direct product of finitely many connected spaces is also connected, then the space of unitary convolutions is connected. $\square$

# D   Analysis on various benchmarked tasks

## D.1   Learning random unitary

In addition to the analysis shown in the main text, here we include a plot showing the value of the Frobenius error as a function of the runtime (Figure 6). Using PROJUNN-T to peform learning via low rank approximations to the gradient significantly speed up learning. Optimization was performed with vanilla gradient descent over batch sizes of 16 with learning rates of 0.5 and 0.33 for PROJUNN-T and PROJUNN-D respectively. Since the learning rate was fixed across

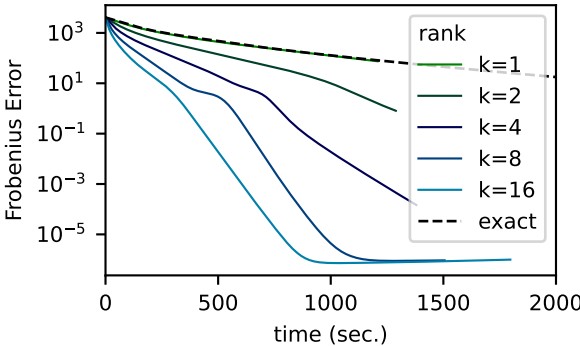

Figure 6: Runtime of PROJUNN-T in learning a random target unitary matrix is faster when using low rank approximations. Here we plot Frobenius error $\|\boldsymbol{U} - \boldsymbol{U}_{tar}\|_F^2$ over the course of optimization. The learning rate is fixed for each value of $k$.

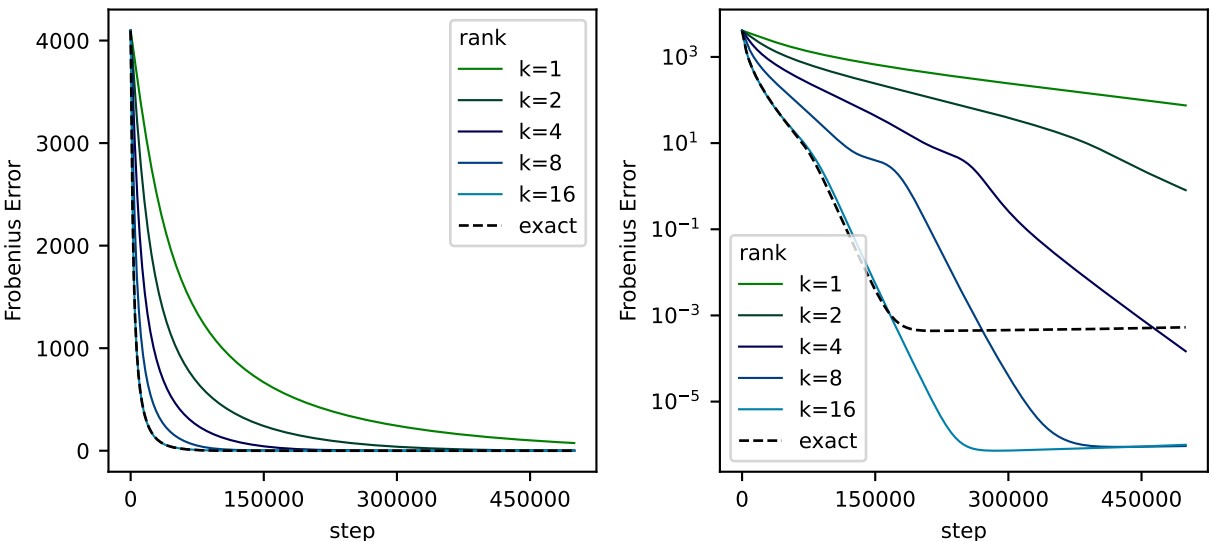

Figure 7: Learning trajectory of PROJUNN-T equipped with the column sampling approximation in the random unitary learning task.

all $k$ for each of these experiments, the norm of the update was smaller for lower values of $k$. Scaling up the learning rate based on the value of $k$ could make runtimes even faster for lower values of $k$.

For sake of completeness, we also include plots (in both regular and logarithmic scaling axis) showing the learning trajectory of the various combinations of samplers and PROJUNN architectures in the random unitary task. See Figure 7 for PROJUNN-T with column sampling (left hand side repeated from main text), Figure 8 for PROJUNN-D with column sampling, Figure 9 for PROJUNN-T with LSI sampling, and Figure 10 for PROJUNN-D with LSI sampling.

Figure 8 and Figure 10 also highlight potential instabilities in training PROJUNN-D over long periods of time, a feature noted in the main text and discussed further in Appendix G.3. To alleviate this, for PROJUNN-D architectures, we set the learning rate to slightly lower at 0.33 and projected parameter matrices onto the closest unitary using Lemma 4.1 every 2048 steps. Note, that since this projection step was performed sparingly every $n$ steps (where $n$ is the dimension of the matrix), we did not find any significant decrease in runtime.

## D.2 Adding task

As shown in the main text, our PROJUNN is able to learn the adding task even for long sequence lengths. Rank $k$ gradient approximations were obtained using the column sampling approximation. RMSprop was used to train

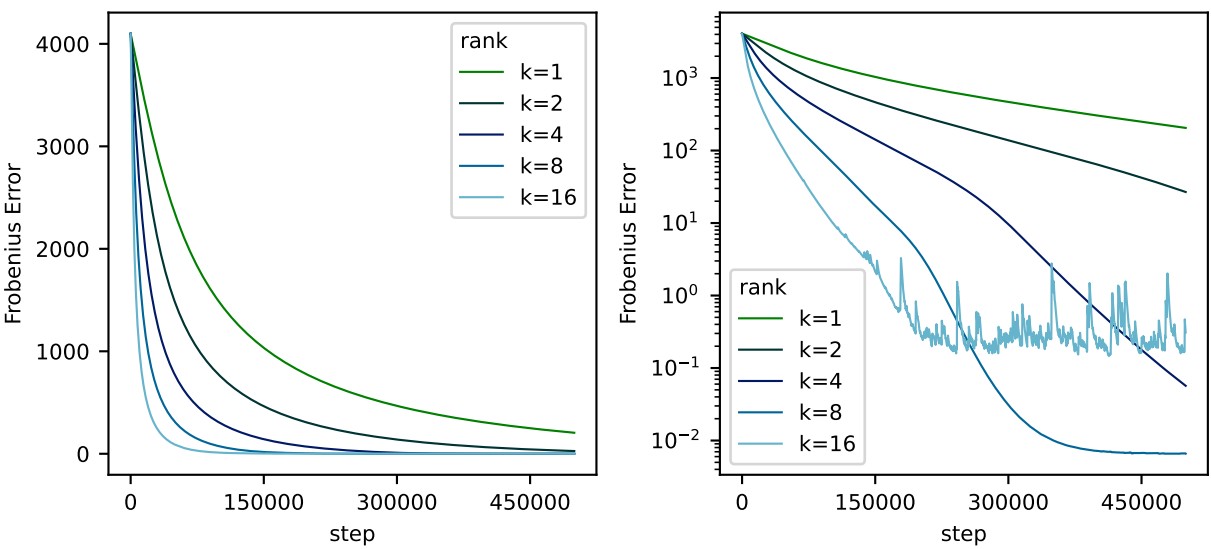

Figure 8: Learning trajectory of PROJUNN-D equipped with the column sampling approximation in the random unitary learning task. Performing gradient updates using exact formulas was too computationally expensive for PROJUNN-D and thus not included here.

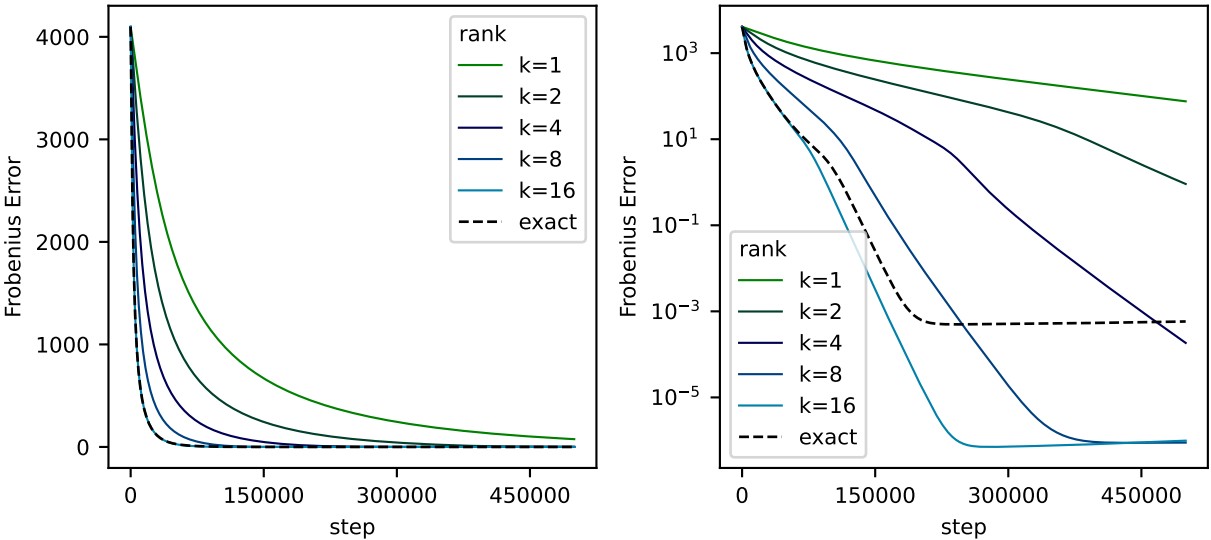

Figure 9: Learning trajectory of PROJUNN-T equipped with the LSI sampling approximation in the random unitary learning task.

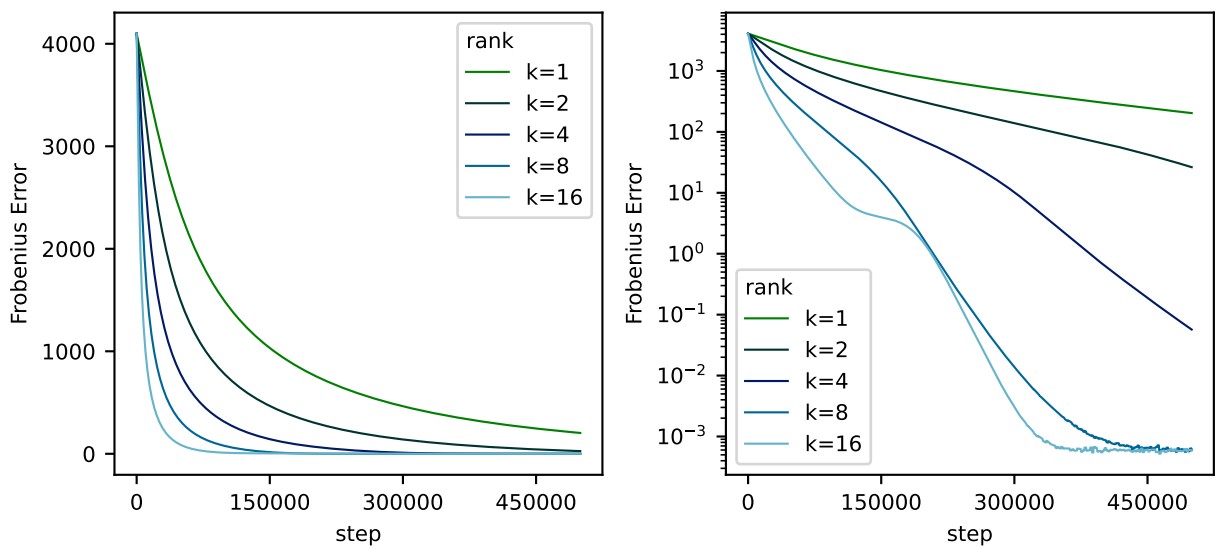

Figure 10: Learning trajectory of PROJUNN-D equipped with the LSI sampling approximation in the random unitary learning task. Performing gradient updates using exact formulas was too computationally expensive for PROJUNN-D and thus not included here.

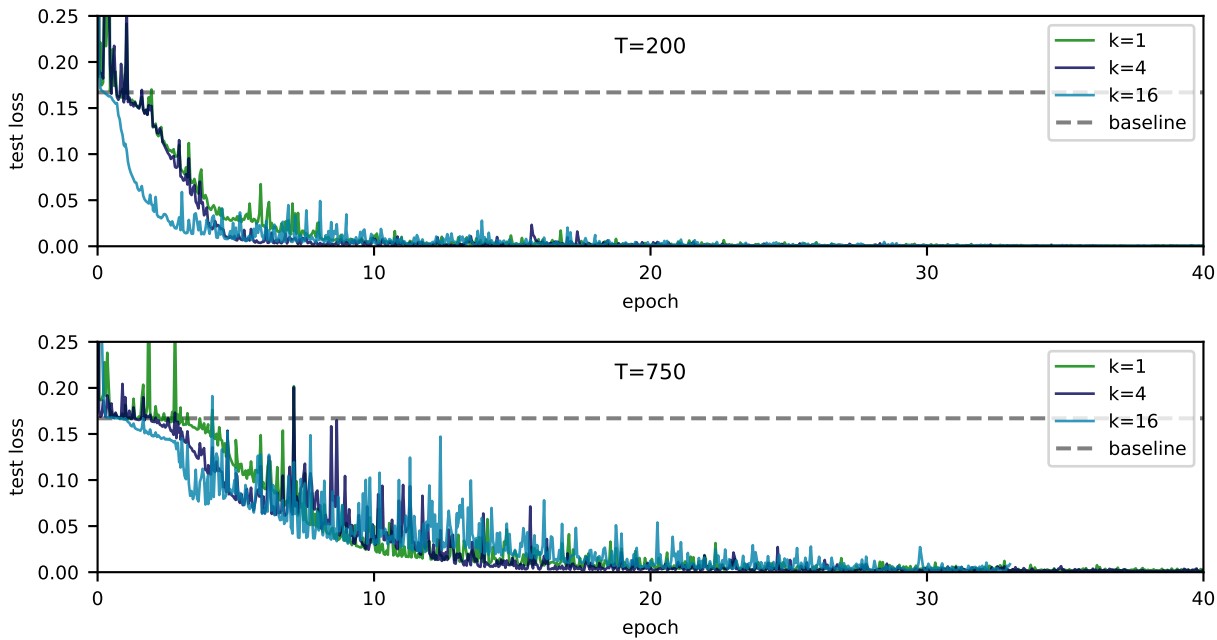

Figure 11: PROJUNN-T (with column sampling approximation) is effective at learning the adding task. Above is a copy of Figure 3 except the plot here is expanded in size and test error is not smoothed.

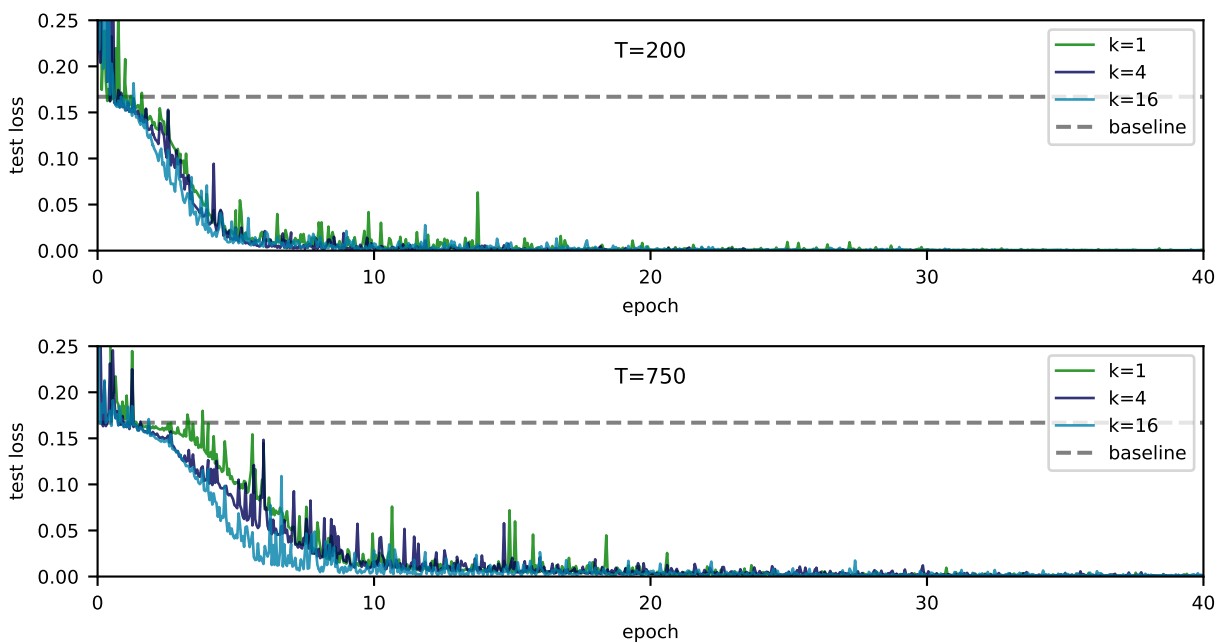

Figure 12: PROJUNN-T (with LSI sampling approximation) is effective at learning the adding task.

Table 3: Test accuracy on CIFAR10 with different unitary convolution parameterizations and our proposed PROJUNN algorithm. We stress that SOC is an approximate unitary parameterization.

| Model | Standard | BCOP | SOC | Cayley | Proj-D | Proj-T |
|---|---|---|---|---|---|---|
| | | | Convolution Type | | | |
| Resnet9 | 92.26 | 80.72 | - | 81.70 | 80.75 | **82.06** |
| Resnet18 | 95.10 | 92.38 | **94.24** | - | 89.43 | 89.59 |

the RNN in this task. The learning rate was initialized to $0.001$ for non-orthogonal parameters and $0.001/32$ for all orthogonal parameters. Each epoch, the learning rate was reduced by multiplying it by $0.96$. We found that initializing the orthogonal matrix as the identity matrix worked well for this task. This is in distinction with *e.g.,* [35] which initialized using the Cayley initialization (block diagonal). For reference, we also include here an enhanced version of the plot in the main text in Figure 11 and a similar plot obtained using PROJUNN with LSI sampling in Figure 12.

### D.3   Copy task

Figure 4 in the main text shows that our PROJUNN-T can efficiently learn the copy task. For this task, we set the learning rate of the RMSprop optimizer to $7e − 4$. For orthogonal parameters, this learning rate was divided by 32. Orthogonal matrices were intialized using Henaff initialization [36] as described in Appendix G.4. We found that this initialization scheme worked best in comparison to other methods.

### D.4   Pixel permuted MNIST

Figure 13 charts the trajectory of learning on the permuted MNIST task. As is evident in the figure, rank $k = 1$ is sufficient to guarantee optimal or nearly optimal convergence in all settings.

### D.5   CIFAR10 CNN experiments

To explore the performance of our PROJUNN training algorithm for convolutional layers, we analyze its performance on CIFAR10 classification. Here, we provide further details to the results in the main text as well as some further preliminary experiments on unitary CNNs in resnet architectures. As in prior work [76] we employ the usual data-

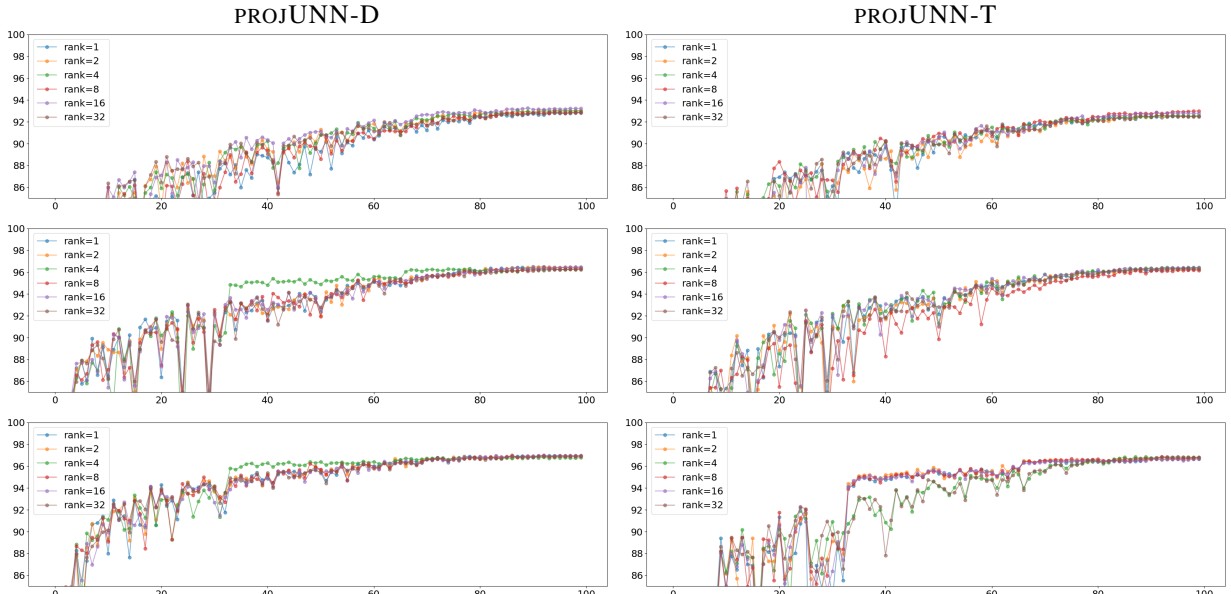

Figure 13: Evolution of the test set accuracy during training at each epoch for the pixel-MNIST task. We depict the evolution for different RNN width (from top to bottom: 116,360 and 512). We observe that regardless of the rank $k$ of the projUNN update, we reach the same final performances.

augmentation of random translations and left-right flips. We leverage the Resnet [34] architecture. As our previous analysis in the RNN setting has shown that rank $k = 1$ is sufficient for convergence, we always set $k = 1$ when using PROJUNN in the convolutional setting. We leverage the RMSprop optimizer and perform cross-validation on the learning rate. We present our results in Table 3 and make two observations. First, the smaller model (Resnet9) is able to reach or slightly outperform the alternative exact orthogonal constraints. Second, the larger model (Resnet18) falls behind the approximate orthogonal constraint method. This result is perhaps expected as our convolutional layers are full-width, *i.e.,* they allow for much greater degree of over-fitting. This was not detrimental in the small model with fewer convolutional layers. As a result, although we validate the ability of PROJUNN to produce state-of-the-art small convolutional networks, there remains open avenues of research to extend the method to larger models. As an aside, the resnet architecture is, in a sense, a very stable architecture since it is precisely designed to be able to incorporate many layers. To provide an honest comparison to prior work, we analyzed the performance of PROJUNN with respect to this resnet architecture, but note that more "vanilla" CNN architectures may be better targets for unitary constraints.

# E    Analysis of low rank updates

It is typically the case that gradients of matrices with respect to a loss function are not *exactly* low rank but *approximately* low rank. More specifically, a matrix $\boldsymbol{A}$ is *approximately* low rank if there exists a rank $k$ matrix $\boldsymbol{A}_k$ such that the relative error of the approximation $E_{rel}$ is small:

$$E_{rel} = \frac{\|\boldsymbol{A} - \boldsymbol{A}_k\|_F}{\|\boldsymbol{A}\|_F} \tag{E.1}$$

where $\|\cdot\|_F$ denotes the Frobenius norm of a matrix. We note that there is a connection between the above and the stable rank of a matrix $\boldsymbol{A}$ defined as $\|\boldsymbol{A}\|_F^2/\|\boldsymbol{A}\|_2^2$. The stable rank is upper bounded by the exact or hard rank of a matrix $\boldsymbol{A}$.

In RNN settings, Figure 14 which plots $1 - E_{rel}$ shows that low rank approximations to a gradient can typically be very close to the true gradient even as the hidden size of the RNN is large. In Figure 14, the RNN was given an input of batch size 64 where each input is a sequence of length 20 and each element of the sequence is a vector in $\mathbb{R}^{200}$ with elements drawn from the standard normal distribution. The RNN outputs logits to learn random target integers ranging from 1 to 10. Gradients were calculated for a single optimization step over a single batch of the random input and output data.

As observed in the main text, such low rank behavior is often more evident in real-world data. In fact, as seen in Figure 1a, virtually all the information of the gradient in a convolutional architecture is captured in the first few singular

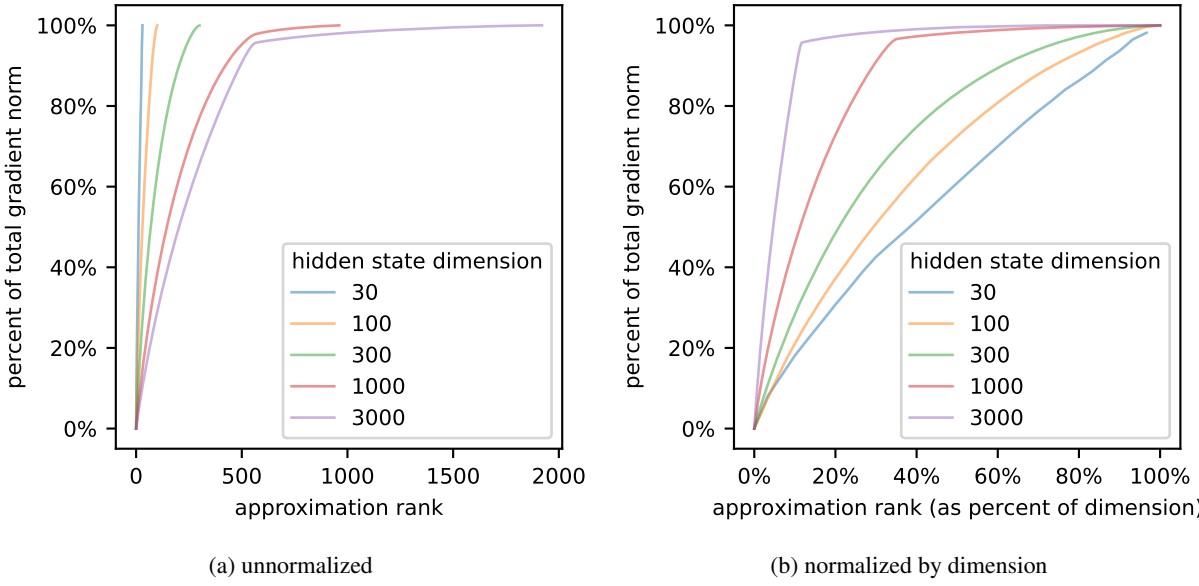

(a) unnormalized                    (b) normalized by dimension

Figure 14: Portion of Frobenius norm (1-$E_{rel}$) captured by a low rank approximation to the gradient of the matrix typically improves with the dimension of the matrix. For larger matrices, at least half of the Frobenius norm of the gradient can be captured with a low rank approximation of a small percent of the overall dimension. Here, we assume that low rank approximations to the matrix are optimal and the RNN is trained on random inputs and outputs. Plots are for the matrix performing transformation from hidden states to hidden states (*i.e.,* the matrix replaced by a unitary/orthogonal matrix in PROJUNN implementations.

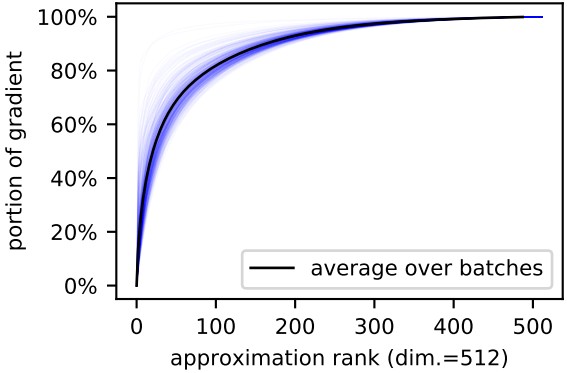

Figure 15: Low rank approximations capture most of the Frobenius norm of the gradient. Here, we plot gradients of the matrix in the RNN's hidden layer for each batch (light blue line) over an epoch of training PROJUNN in the pixel-by-pixel MNIST task (see Section 5).

vectors. Gradients here are shown for a single $C \times C$ block in the Fourier regime of the convolution filter parameterized via our orthogonal PROJUNN convolution (see Section 4.4 and Appendix B.1 for form of parameterization). This filter is contained in the last residual block of the Resnet-9 network [34] and has $512$ channels so the gradient is a $512 \times 512$ matrix. Since networks were trained with a batch size of 128, the maximum rank of this gradient is actually 128 (see short proof below in Proposition E.2).

Since the weight matrix in hidden layers of RNNs is repetitively applied, gradients tend to be of a higher stable rank in these settings. Nevertheless, as evident in Figure 15, gradients over the course of an epoch of training an RNN of hidden dimension 512 on the sequential MNIST task still exhibit clear low rank behavior, albeit not to the same degree as convolutional architectures. In constructing these plots, a batch size of 128 was used and networks were trained using the RMSprop algorithm. Throughout our experience with training the PROJUNN on various tasks, we observe that setting the rank of an approximation to even just one is effective at ensuring convergence of training to a good solution.

We furthermore note that the exact rank of a gradient in neural network architectures can typically be bounded by the dimensions of the input. For convolutional networks, this exact rank is bounded by the batch size and for vanilla recurrent neural networks, this exact rank is bounded by the batch size times the sequence length. We provide formal statements and short proofs of these propositions below.

**Proposition E.1.** *Given a loss function $\ell : \mathbb{R} \times \mathbb{R} \to \mathbb{R}$ taking in two real numbers and outputting a real number, let $f^{(t)}$ denote the $t$-th sequential output of a vanilla RNN defined as*

$$\begin{aligned}
\boldsymbol{h}^{(t)}(\boldsymbol{x}) &= \sigma(\boldsymbol{M}\boldsymbol{x}^{(t)} + \boldsymbol{W}\boldsymbol{h}^{(t-1)}(\boldsymbol{x})) \\
f^{(t)}(\boldsymbol{x}) &= \tau(\boldsymbol{V}\boldsymbol{h}^{(t)}(\boldsymbol{x})),
\end{aligned} \tag{E.2}$$

*where $t \in [T]$ indexes the sequence of inputs, inputs $\boldsymbol{x}^{(t)} \in \mathbb{R}^d$ ($\boldsymbol{x}$ denotes the concatenation of all inputs), input transformation $\boldsymbol{M} \in \mathbb{R}^{h \times d}$, hidden transformation $\boldsymbol{W} \in \mathbb{R}^{h \times h}$, output transformation $\boldsymbol{V} \in \mathbb{R}^{h \times d}$, and $\sigma$ and $\tau$ are pointwise non-linearities. Let $\nabla_{\boldsymbol{W}} L$ indicate the gradient of the hidden transformation matrix $\boldsymbol{W}$ with respect to the loss function $L = \sum_{i=1}^{b} \ell(f^{(T)}(\boldsymbol{x}_i), y_i)$ over a batch of inputs $\{\boldsymbol{x}_i, y_i\}_{i=1}^{b}$. The rank of $\nabla_{\boldsymbol{W}} L$ is bounded above by $bT$.*

*Proof.* Define $\boldsymbol{z}^{(t)} = \boldsymbol{M}\mathbf{x}^{(t)} + \boldsymbol{W}\boldsymbol{h}^{(t-1)}(\boldsymbol{x})$. Calculating the gradient of the loss function with respect to the parameter $\boldsymbol{W}$, we have:

$$\begin{aligned}
\nabla_{\boldsymbol{W}} L &= \sum_{i=1}^{b} \nabla_{\boldsymbol{W}} \ell(f^{(T)}(\boldsymbol{x}_i), y_i) \\
&= \sum_{i=1}^{b} \sum_{t=1}^{T} \sum_{k=1}^{h} \left[ \frac{\partial \ell(f^{(T)}(\boldsymbol{x}_i), y_i)}{\partial \boldsymbol{z}^{(t)}} \right]_k \frac{\partial \boldsymbol{z}_k^{(t)}}{\partial \boldsymbol{W}} \\
&= \sum_{i=1}^{b} \sum_{t=1}^{T} \left[ \frac{\partial \ell(f^{(T)}(\boldsymbol{x}_i), y_i)}{\partial \boldsymbol{z}^{(t)}} \right] \left[ \boldsymbol{h}^{(t-1)}(\boldsymbol{x}_i) \right]^{\mathsf{T}}.
\end{aligned} \tag{E.3}$$

The above is the sum of $bT$ rank one matrices which is at most rank $bT$ concluding the proof. $\qquad \square$

**Proposition E.2.** *Given a loss function $\ell : \mathbb{R} \times \mathbb{R} \to \mathbb{R}$ taking in two real numbers and outputting a real number and convolution filter $\mathbf{W} \in \mathbb{R}^{N \times M \times C \times C}$, let $f(\mathbf{X})$ denote the convolutional neural network defined as*

$$\begin{aligned}
\mathbf{H}(\mathbf{X}) &= g_{pre}(\mathbf{X}) \\
\mathbf{Y}(\mathbf{X}) &= \mathrm{conv}_{\mathbf{W}}(\mathbf{H}(\mathbf{X})) \\
f(\mathbf{X}) &= g_{post}(\mathbf{Y}(\mathbf{X})),
\end{aligned} \tag{E.4}$$

*where input $\mathbf{X} \in \mathbb{R}^{N_{in} \times M_{in} \times C_{in}}$ and $g_{pre}$ and $g_{post}$ are functions which apply the layers before and after the convolution with filter $\mathbf{W}$. Let $\widehat{\mathbf{W}}_{\widehat{r}, \widehat{s}, :, :}$ denote the $C \times C$ matrix storing the values of the convolution filter in the Fourier regime for frequencies $\widehat{r}$ and $\widehat{s}$ (see Equation (B.15)). Let $\nabla_{\widehat{\mathbf{W}}_{\widehat{r}, \widehat{s}, :, :}} L$ indicate the gradient of $\widehat{\mathbf{W}}_{\widehat{r}, \widehat{s}, :, :}$ with respect to the loss function $L = \sum_{i=1}^{b} \ell(f(\mathbf{X}_i), y_i)$ over a batch of inputs $\{\mathbf{X}_i, y_i\}_{i=1}^{b}$. Then, for all $\widehat{r}$ and $\widehat{s}$ in the support of the filter, the rank of $\nabla_{\widehat{\mathbf{W}}_{\widehat{r}, \widehat{s}, :, :}} L$ is bounded above by $b$.*

*Proof.* We have that

$$\nabla_{\widehat{\mathbf{W}}_{\widehat{r}, \widehat{s}, :, :}} L = \sum_{i=1}^{b} \nabla_{\widehat{\mathbf{W}}_{\widehat{r}, \widehat{s}, :, :}} \ell(f(\mathbf{X}_i), y_i). \tag{E.5}$$

We now proceed to show that the rank of $\nabla_{\widehat{\mathbf{W}}_{:, :, \widehat{r}, \widehat{s}}} \ell(f(\mathbf{X}_i), y_i)$ is equal to one which completes the proof. As a reminder, we have via the convolution theorem that

$$[\mathrm{FFT}\, \mathrm{conv}_{\mathbf{W}}(\mathbf{X})]_{\widehat{r}, \widehat{s}, :} = \widehat{\mathbf{W}}_{\widehat{r}, \widehat{s}, :, :}^{*}\, [\mathrm{FFT}\, \mathbf{X}]_{\widehat{r}, \widehat{s}, :}. \tag{E.6}$$

Let $\widehat{\mathbf{Y}}(\mathbf{X}) = \mathrm{FFT}\, \mathrm{conv}_{\mathbf{W}}(\mathbf{X})$, then we have

$$\begin{aligned}
\nabla_{\widehat{\mathbf{W}}_{:, :, \widehat{r}, \widehat{s}}} \ell(f(\mathbf{X}_i), y_i) &= \sum_{k=1}^{C} \frac{\partial \ell(f(\mathbf{X}_i), y_i)}{\partial \widehat{\mathbf{Y}}(\mathbf{X})_{\widehat{r}, \widehat{s}, k}} \frac{\partial \widehat{\mathbf{Y}}(\mathbf{X})_{\widehat{r}, \widehat{s}, k}}{\mathbf{W}_{\widehat{r}, \widehat{s}, :, :}} \\
&= \frac{\partial \ell(f(\mathbf{X}_i), y_i)}{\partial \widehat{\mathbf{Y}}(\mathbf{X})_{\widehat{r}, \widehat{s}, :}} \left( [\mathrm{FFT}\, \mathbf{X}]_{\widehat{r}, \widehat{s}, :} \right)^{\mathsf{T}}.
\end{aligned} \tag{E.7}$$

| Model | Complexity of forward + backward step | Notes |
|---|---|---|
| BCOP [1] | $O(C^2W^2N^2 + C^3W^3)$ | $W$ sequential orthogonalizations needed (slow for large $W$) |
| SOC [2] | $O(pC^2W^2N^2)^a$ | $p$ denotes number of terms in Taylor series approximation |
| Cayley [3] | $O(N^2C^2\log(N) + N^2C^3)^b$ | |
| PROJUNN (our method) | $O(N^2C\log(N) + kN^2C^2)^c$ | $k$ denotes rank of gradient updates |

[1] [58], [2] [73], [3] [76]

[a] just applying the convolution (without gradient update) also requires the added factor $p$ in runtime unlike standard convolutions and BCOP which run in time $O(C^2W^2N^2)$ upon just evaluation, [b] approximations to matrix inversion exist which may reduce runtimes though these approximations are not implemented here, [c] runtime shown for typical setting when $k \ll n$

Table 4: Time complexity of 2-D orthogonal convolutional layers with filter size $W \times W$ applied to $N \times N$ inputs with $C$ input and output channels.

The above is a rank one matrix which concludes the proof. $\square$

### E.1 Other sampling algorithms

The prior analysis showed that gradients are typically low rank, and in the main text, we listed a couple efficient algorithms that allow one to approximate a full but approximately low rank matrix with an explicitly low rank matrix. Enhancements to these algorithms exist that may provide further improvements in very large dimensions. For example, the *FKV sampling* [27] and *constant time SVD* [23] algorithms provide runtimes that are often logarithmic in the dimension of the matrix for sampling entries from the low rank approximation to a matrix. Nevertheless, explicitly constructing all entries of a rank $k$ approximation to an $n \times n$ matrix $\boldsymbol{A}$ requires time at least $O(k^2n)$. Improvements in runtime are achieved by applying random projections to both the column and row subspaces of the matrix $\boldsymbol{A}$ to perform the final approximation. In practice, the runtime of these algorithms depends on other factors that limit the applicability of the algorithm even for relatively large matrices. Numerical simulations show that FKV sampling achieves a practical speedup for matrices of dimension approximately $10^6$ or higher [8]. Since matrices in our PROJUNN were of much smaller dimension, we did not use these methods.

## F Runtime comparisons

Table 1 (see main text) and Table 4 lists the asymptotic runtime complexities of unitary/orthogonal models in the RNN and convolutional setting respectively. For RNNs, PROJUNN has the best runtime scaling of all models which parameterize the full orthogonal/unitary space with just one layer. In fact, apart from the rank approximation factor $k$, the runtime of PROJUNN is optimal in the RNN setting. In the convolutional setting, PROJUNN scales most efficiently when there are many channels or the filter size $W$ scales faster than the logarithm of the input dimension $\log N$. For small filter sizes, BCOP and SOC scale relatively efficiently. In fact, SOC is nearly optimal for $W \ll \log N$ up to the approximation factor $p$ in the Taylor expansion which is required both upon evaluation and training of the network.

**Runtimes for RNN implementations in practice** Enforcing unitarity in PROJUNN incurs a computational overhead associated to performing eigendecompositions and QR decompositions entailed in updating the gradient. Even though such computations are performed on a small subspace of the total dimension of the matrix, such computations may not effect increase training times by a constant factor as evident in Figure 2a. Empirically, we observed that much of the increased overhead was due to performing eigendecompositions and QR decompositions on a GPU, both tasks which are challenging to parallelize on GPU architectures. We note that similar issues may be one reason why the expRNN runtimes appear much slower in our simulations as shown in Figure 2a. Recent updates to pytorch [70] and tensorflow [2] have focused on improving the runtimes of these common linear algebraic routines, and we expect these runtimes to continue to improve in the future.

**Runtimes for convolutional network implementations in practice** Figure 16, Figure 17, and Figure 18 compare runtimes of orthogonal convolution algorithms when varying the number of channels, input size, and filter size respectively. Figure 19 plots the runtime when the filter and image size are set equal to each other and varied together. In summary, we find that the skew orthogonal convolution (SOC) [73], the Cayley model [76], and our PROJUNN are all relatively efficient for small to medium sized models. PROJUNN empirically performs best in comparison to other models when there are many channels or the filter size is large. SOC [73] is the fastest when the filter size is small. As a reminder, SOC [73] employs the Taylor series approximation to form a unitary/orthogonal matrix. We did not change

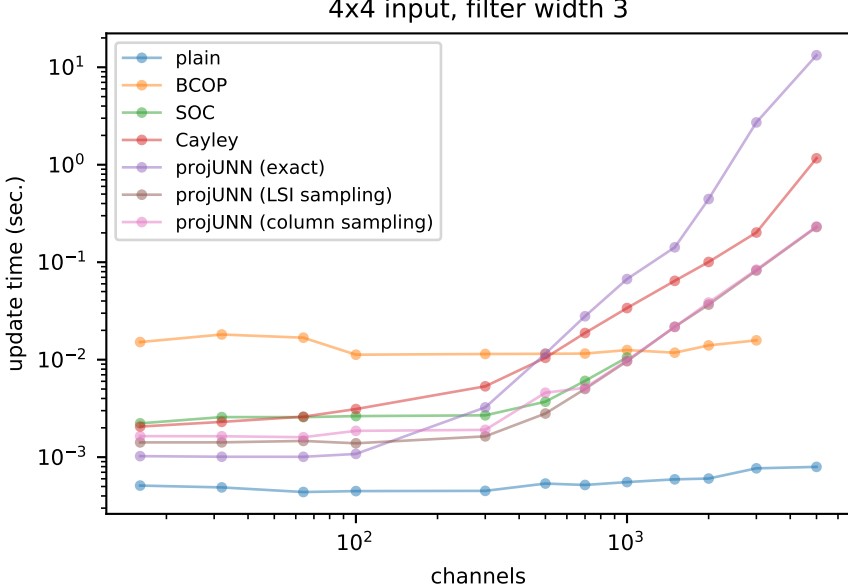

Figure 16: Comparison of runtimes of orthogonal convolutional architectures when varying the number of channels (log-log plot). The number of input channels is equal to the number of output channels. Runtimes are averaged over 10 forward and backward passes through a single convolutional operation on random data. PROJUNN with LSI sampling and column sampling have very similar runtimes and may appear to completely overlap on the chart. Out of memory error obtained for BCOP for large number of channels.

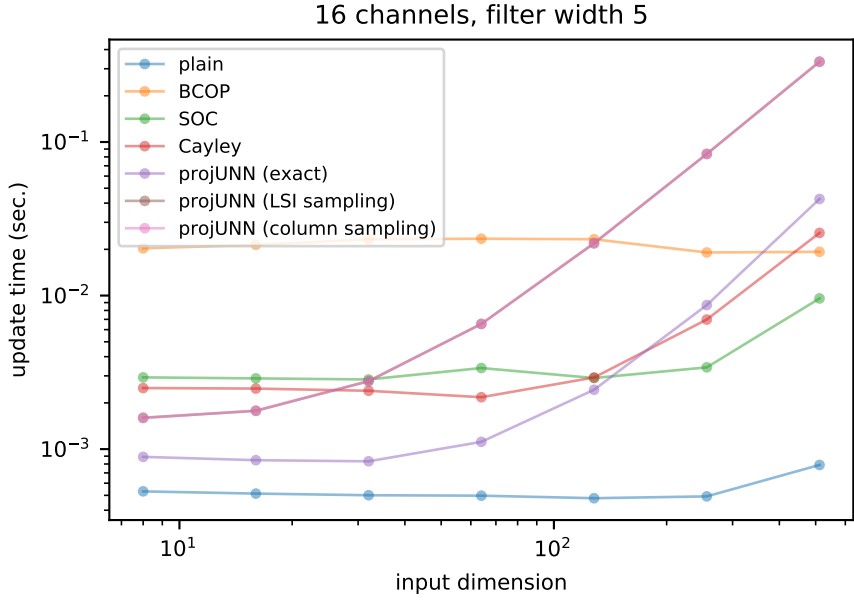

Figure 17: Comparison of runtimes of orthogonal convolutional architectures when varying the size of the input which is a square image (log-log plot). Runtimes are averaged over 10 forward and backward passes through a single convolutional operation on random data. PROJUNN with LSI sampling and column sampling have very similar runtimes and may appear to completely overlap on the chart.

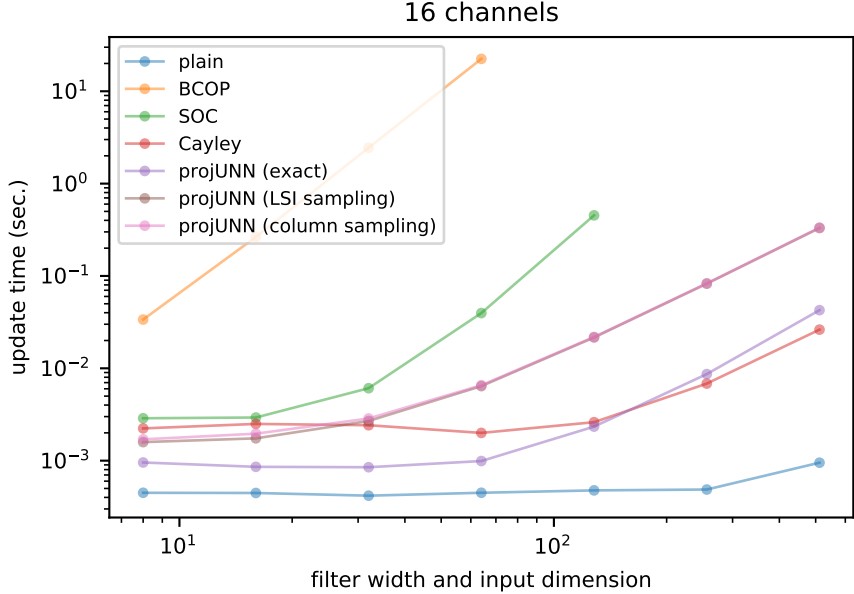

Figure 18: Comparison of runtimes of orthogonal convolutional architectures when varying the size of the filter (log-log plot). Both PROJUNN and the cayley convolution [76] perform operations on the full space of convolution filters so the filter size does not change the runtime for these models. Runtimes are averaged over 10 forward and backward passes through a single convolutional operation on random data. PROJUNN with LSI sampling and column sampling have very similar runtimes and may appear to completely overlap on the chart. Out of memory error obtained for SOC beyond filter size of 31.

Figure 19: Comparison of runtimes of orthogonal convolutional architectures when varying the size of the filter and the input dimension in-tandem (log-log plot). Runtimes are averaged over 10 forward and backward passes through a single convolutional operation on random data. PROJUNN with LSI sampling and column sampling have very similar runtimes and may appear to completely overlap on the chart. Out of memory error obtained for SOC beyond filter size and image size of 128.

the number of terms in the approximation used even though increasing the size of a filter, dimension of an input, or the number of channels will likely require more terms to maintain the same error in the approximation.

The Cayley model [76] has a runtime that is empirically similar to PROJUNN in most of our experiments despite PROJUNN having an improved asymptotic scaling with regards to the number of channels (see Table 4). Runtimes here are calculated by averaging across the first 10 steps of gradient descent on random data for initialized networks. Close to initialization, matrices in the convolution filter are generally well conditioned and this may be one reason why the Cayley model showed good scaling in Figure 16 since the matrix inversion for the well conditioned matrices runs quickly. We also suspect that for the number of channels considered, the added costs of performing sampling and projections in PROJUNN do not materialize into a very clear runtime benefit. Interestingly, the exact version of PROJUNN (no low rank approximation) runs in similar time to the Cayley method.

Finally, we note that in comparison to RNN implementations, there is increased overhead associated to performing eigendecompositions and QR decompositions with our projUNN convolutional network implementations since unitary matrices are batched across the elements of the convolutional filter. As mentioned before, much of the increased overhead was due to performing eigendecompositions and QR decompositions on a GPU, both tasks which can be challenging to parallelize on GPU based architectures. Future updates to Pytorch [70] and Tensorflow [2] may help improve runtimes of these operations.

# G    Network architectures and training details

| ExpRNN trivializations | https://github.com/Lezcano/expRNN | MIT |
| scoRNN | https://github.com/SpartinStuff/scoRNN | NA |
| scuRNN | https://github.com/Gayan225/scuRNN | NA |
| skewed orthogonal convolutions (SOC) | https://github.com/singlasahil14/SOC | NA |
| Cayley convolution | https://github.com/locuslab/orthogonal-convolutions | MIT |
| BCOP | github.com/ColinQiyangLi/LConvNet | MIT |

## G.1    Handling complex numbers

Since matrices in the unitary group are complex valued, care must be taken to ensure that the neural network can handle such complex numbers. In these settings, standard activation functions need to be adapted for complex numbers. It is often advantageous to have a nonlinearity which does not change the phase of its input. For these reasons, the activation function used is typically the modRELU activation shown below:

$$\sigma_{modRELU}(z) = (|z| + b)\frac{z}{|z|} \ \ \text{if} \ \ \|z\| + b > 0$$
$$\sigma_{modRELU}(z) = 0 \ \ \text{if} \ \ |z| + b \le 0$$

(G.1)

where $b$ is a bias term (trainable). In calculations of the modulus $|z|$, a small additive constant is often added to avoid stability issues when $|z|$ is small.

Another challenge that arises with performing orthogonal convolution is that the representation of a real-valued convolutional filter in the Fourier domain will be complex-valued. More specifically, the fast Fourier transform of a real signal is Hermitian-symmetric. For example, for a one dimensional vector $x \in \mathbb{R}^N$ where $\hat{x} = \text{FFT}\, x \in \mathbb{C}^N$ and $\hat{x}_i = \hat{x}_{-i \mod N}$. Therefore, when initializing weight filters in our orthogonal convolution operations, one must be careful to ensure the Hermitian symmetric property holds. Similarly, when performing convolutions in the Fourier space, care must be taken in converting data types to and from complex and real space. In our implementation, we used the pytorch FFT.RFFT2 and FFT.IRFFT2 commands to implement these operations efficiently [69].

## G.2    Optimizers

For training PROJUNN architectures, a separate optimizer is used for unitary/orthogonal parameters which must be projected after updates and all other parameters. Learning rates for the unitary/orthogonal parameters are typically set to less than that of the other parameters. In our implementations, we found that a learning rate for the unitary or orthogonal parameters of one-tenth or one-twentieth of that of the other parameters works well in practice. Furthermore, when using optimizers with momentum terms, we constructed a variant of standard optimizers to apply changes to the momentum terms after projecting gradients onto the unitary/orthogonal manifold or tangent space. This optimizer performed well in our experiments though sometimes added instability. Therefore, unless otherwise stated, standard optimizers were used.

### G.3 Numerical stability

Though updates via PROJUNN mathematically guarantee that parameters remain unitary/orthogonal, performing a significant number of sequential updates to a matrix can add numerical errors over time slowly drifting parameters away from the unitary/orthogonal manifold. This is especially true in the case of PROJUNN-D where updating matrices requires division of eigenvalues in the eigendecompositon of the rank $k$ subspace (see Appendix C and Equation (C.16)). In contrast, we empirically find that PROJUNN-T maintains stability throughout optimization even when requiring many updates. To actively address potential instabilities, one can perform eigendecompositions with higher floating point precision or sporadically project unitary/orthogonal parameters onto the closest unitary/orthogonal matrix via Lemma 4.1. Though this projection step requires $O(n^3)$ time for an $n \times n$ matrix, performing this projection only every $O(n)$ gradient updates can still preserve efficient runtimes of on average $O(kn^2)$ time per gradient update. Unless otherwise stated, we do not perform any additional steps for maintaining stability.

### G.4 Initialization of unitary/orthogonal parameters

Empirically, we found that initializing unitary/orthogonal matrices to be Haar random or close to Haar random does not perform well. This is also an observation noted in prior works [36, 35, 56]. Instead, we used one of two different initialization schemes. The first initializes unitary/orthogonal parameters as the identity matrix. The second initializes unitary/orthogonal matrices using variants of the so-called Henaff initialization [36] where $2 \times 2$ diagonal blocks of the matrices are initialized as samples from the below

$$\exp\left( \begin{bmatrix} 0 & s_i \\ -s_i & 0 \end{bmatrix} \right),$$
(G.2)

where $s_i$ is sampled uniformly from $[-\pi, \pi]$ [36]. In other words, an $n \times n$ matrix $\boldsymbol{U}$ is initialized as

$$\boldsymbol{U} = \begin{bmatrix} \exp\left( \begin{bmatrix} 0 & s_1 \\ -s_1 & 0 \end{bmatrix} \right) & \boldsymbol{0} & \cdots & \boldsymbol{0} \\ \boldsymbol{0} & \exp\left( \begin{bmatrix} 0 & s_2 \\ -s_2 & 0 \end{bmatrix} \right) & & \boldsymbol{0} \\ \vdots & & \ddots & \vdots \\ \boldsymbol{0} & \boldsymbol{0} & \cdots & \exp\left( \begin{bmatrix} 0 & s_{n/2} \\ -s_{n/2} & 0 \end{bmatrix} \right) \end{bmatrix}.$$
(G.3)

Note, that since we parameterize matrices in the Lie group instead of the Lie algebra, we include the exponential map in the parameterization above. Other variants of the above have been used in prior works. For example, the Cayley initialization samples $s_i = \sqrt{\frac{1-\cos t_i}{1+\cos t_i}}$ where $t_i$ is sampled uniformly from $[0, \pi/2]$ [35].

### G.5 Computational details

We employed the latest version of PyTorch (1.10.1+cu102) with Python3.8 and all the pre-requisite libraries coming along. The hardware leverages Quadro GP100 GPU and Intel(R) Xeon(R) CPU E5-2698 v4 @ 2.20GHz.