# OpenReview forum: "projUNN: efficient method for training deep networks with unitary matrices"
_NeurIPS.cc/2022/Conference — NeurIPS 2022 Accept_

### Official Review · Reviewer_o9su · 2022-07-09

**Rating:** 8
**Confidence:** 3
**Soundness:** 4 excellent
**Presentation:** 4 excellent
**Contribution:** 3 good

**Summary:**

The authors propose two optimization algorithms for training deep networks with unitary weight matrices. Then they extend their approach to unitary convolutional operators. Unlike existing approaches, they exploit the low-rank structure of gradients that is a consequence of the relatively small batch sizes used in SGD-like optimization. This way they achieve significant speedups over existing unitary training approaches.

**Questions:**

-What are applications beside the RNN toy problems where unitary operators may be needed?


**Limitations:**

The authors have adequately addressed the limitations. I cannot think of any potential negative societal impact

**Strengths And Weaknesses:**

Strength
- The authors propose two novel projected gradient step procedures for training constraint to the space of unitary matrices. In contrast to exisitng approaches, we achieve more computational efficiency by exploiting the low-rank structure of gradients that is present in minibatch training.
- Experimental evaluations show that the low-rank approximation even allows ranks substantially smaller than the batch size ( times sequence length for RNN), increasing the computational complexity even more.
- Discussion of related work places this work among existing work, clearly stating advances and differences.
- The extension to convolutional layers by exploiting the relation between the Fourier transformation and the discrete convolution operation enables the use of unitary operators in the image domain. This may allow users more control about properties of the linear operators that have been linked to generalization in deep networks like spectral norm, stable rank or eigenvalue decay.
- Source code for recurrent networks is available which facilitates widespread use. Unfortunately the CNN code was not available in time for the review

Weaknesses
- The experiments mostly show technical advantages like increases in computational performance, not benefits in applications where this technique is strictly required

---

> ### Author Response · Authors · 2022-08-01
> **Response to reviewer o9su**
>
> We thank the reviewer for their feedback and their positive comments about our work.
>
> **Reviewer**: “Unfortunately the CNN code was not available in time for the review” \
> **Response**: For reviews, we are including an example simulation of the unitary CNN in an anonymous Google Colab (https://colab.research.google.com/drive/1FCfRKxrbhelP1mcLuiq6PxnCt1BG0wNp?usp=sharing) where we learn MNIST data. This illustrates the use of projUNN in a simple CNN setting. Our full code base will be released as a github link upon publication.
>
> **Reviewer**: “What are applications beside the RNN toy problems where unitary operators may be needed?”\
> **Response**: We agree with the reviewer that it would be useful to find further applications where strictly enforcing unitarity/orthogonality enhances performance for practical applications. Looking beyond the RNN experiments performed here, possible avenues of exploration include enforcing isometry as an inductive bias of the architecture (e.g., see [1] for inspiration), constructing models whose layers are more robust to adversarial attacks (prior works explored this in [2],[3] among others), developing stable invertible layers for normalizing flows (e.g., see [4]), and potentially learning quantum data [5]. In many practical settings, simply regularizing weights to be close to unitary can be effective, so we are searching for settings where strict unitarity is actually necessary.
>
>
> **References**\
> [1] Baddoo, P. J., Herrmann, B., McKeon, B. J., Kutz, J. N., & Brunton, S. L. (2021). Physics-informed dynamic mode decomposition (piDMD). arXiv preprint arXiv:2112.04307.\
> [2] Singla, S., & Feizi, S. (2021, July). Skew orthogonal convolutions. In International Conference on Machine Learning (pp. 9756-9766). PMLR.\
> [3] Trockman, A., & Kolter, J. Z. (2021). Orthogonalizing convolutional layers with the cayley transform. arXiv preprint arXiv:2104.07167.\
> [4] Golinski, A., Lezcano-Casado, M., & Rainforth, T. (2019). Improving normalizing flows via better orthogonal parameterizations. In ICML Workshop on Invertible Neural Networks and Normalizing Flows. \
> [5] Biamonte, J., Wittek, P., Pancotti, N. et al. Quantum machine learning. Nature 549, 195–202 (2017). https://doi.org/10.1038/nature23474

---

### Official Review · Reviewer_Kc6Q · 2022-07-13

**Rating:** 9
**Confidence:** 4
**Soundness:** 4 excellent
**Presentation:** 3 good
**Contribution:** 4 excellent

**Summary:**

The paper present two efficient update rules for neural networks with unitary matrices parametrized as full n x n arrays. Unitary matrices have attractive properties for neural networks since they preserve the norm of vectors they are multiplied to, and unlike orthogonal matrices, they have differentiable parametrizations though the main drawback to their use is that full parametrizations and/or gradient updates are often inefficient ( O(n^3) as opposed to O(n^2) of arbitrary matrices ). This paper proposes efficient O(k n^2) gradient updates, where k is a gradient rank hyperparameter. The methods are based on the hypothesis that the training gradient w.r.t. the unitary matrix can be well approximated as a low-rank matrix. They propose to use existing randomized SVD methods to compute the low-rank approximation

The first method PROJUNN-D first computes a low-rank approximation of the gradient, then it applies a standard SGD step to the unitary matrix, and projects the result back to the unitary group, which can be done efficiently by exploiting the fact that the perturbation was low-rank. The author comments that this method can be subject to numeric instability.

The second method PROJUNN-T first computes a low-rank approximation of the gradient, then it rotates the unitary matrix along a projection of the approximated gradient on the unitary Lie algebra.

PROJUNN-D and PROJUNN-T are evaluated on learning a random unitary matrix and, as a URNN update rule, on various standard RNN benchmarks, with competitive results and a substantial reduction of runtime.

An extension to convolution is also discussed, but it is not shown to be competitive to vanilla convolution.


**Questions:**

None

**Strengths And Weaknesses:**

Strengths:
- Very compelling result, might enable making URNNs practical
- Very good exposition
- Extensive experiments

Weaknesses:
- Both methods seem fairly complicated to implement, full peudocode algorithm descriptions (including the low-rank gradient approximation) would be appreciated. Hopefully code will be released.

---

> ### Author Response · Authors · 2022-08-01
> **Response to reviewer Kc6Q**
>
> We thank the reviewer for their feedback and nice comments about the exposition and presentation of our paper.
>
> We agree with the reviewer that pseudocode would be helpful and we have added this in Appendix D. Space-permitting given the added page in the full publication, we will use the remaining space to add the pseudo-code to the main text.
>
> Furthermore,  we have included another google colab link (https://colab.research.google.com/drive/1FCfRKxrbhelP1mcLuiq6PxnCt1BG0wNp?usp=sharing) which has an example of projUNN used in a convolutional network trained on MNIST. Our complete code base will be released as a github link upon publication of the manuscript.

---

### Official Review · Reviewer_3SuC · 2022-07-21

**Rating:** 7
**Confidence:** 3
**Soundness:** 3 good
**Presentation:** 4 excellent
**Contribution:** 3 good

**Summary:**

The paper studies the acceleration of unitary neural network training. By exploiting the low-rank structure of gradients (done with sampling), the authors design two efficient update methods of unitary matrices based on gradient descent: one that directly projects the updated weight to the unitary manifold, and one that projects the gradient to the tangent space followed by a rotation mapping. Experimental results on several RNN tasks show that the proposed algorithm is on par with existing unitary neural networks while achieving nearly optimal runtime complexity. The authors also perform some preliminary study on training orthogonal convolution layers.

**Questions:**

The authors propose two methods (projUNN-D and projUNN-T). Is there any difference between these two methods other than the numerical issues? I would like to see some comparisons (e.g., the pros and cons) between them.

**Limitations:**

Yes. The authors discussed the numerical instability of the direct method as well as the memory concern when applied to orthogonal convolutions.

**Strengths And Weaknesses:**

The paper is well-motivated and clearly-written. The proposed methods are simple-yet-effective, and the theoretical results are strong (which I have checked in details). Comparison with prior works looks comprehensive. Due to my limited knowledge in related works, I will leave the evaluation of novelty as well as experimental results to other reviewers.

---

> ### Author Response · Authors · 2022-08-01
> **Response to reviewer 3SuC**
>
> We thank the reviewer for their feedback and for their overall positive assessment of our work. We have copied their question below followed by our answer.
>
> **Reviewer**: “The authors propose two methods (projUNN-D and projUNN-T). Is there any difference between these two methods other than the numerical issues?”\
> **Response**: As projUNN-D and projUNN-T are both first-order gradient based methods, their performance is very similar. However, projUNN-D closely follows methods of projected gradient descent whereas projUNN-T follows methods of Riemannian gradient descent. They both agree up to first order (i.e. are essentially equivalent for infinitesimally small learning rate) and are a retraction of the exponential map. Generally, we have found that Riemannian gradient descent methods are more stable and faster as stated in the paper, but we also find projUNN-D to be effective in many cases and also potentially simpler to implement. We hope future work can be inspired by either of these methods to extend the speed-ups found in unitary/orthogonal matrices to other matrix manifolds.

---

### Official Review · Reviewer_BcYz · 2022-07-24

**Rating:** 6
**Confidence:** 3
**Soundness:** 3 good
**Presentation:** 4 excellent
**Contribution:** 3 good

**Summary:**

This paper proposes a method (projUNN) for efficiently training neural networks with parameters that are constrained to be unitary/orthogonal matrices.

The main strategy is to obtain a low rank approximation of the gradient update and then use Riemannian gradient descent either via retraction (method projUNN-D) or exact computation of the exponential map (method projUNN-T).


**Questions:**


It would be interesting to compare train/test loss evolution over iterations between projUNN and the other baselines (i.e. testing convergence in terms of iterations rather than time) as well as total time to achieve similar loss (as opposed to single iteration cost).


Minor Remarks:
Typo: pg. 23 $$\hat a = \tilde{M}^{\dagger} a \rightarrow \hat a = U^{\dagger} a$$



**Limitations:**

Yes

**Strengths And Weaknesses:**

Overall, the paper is clear and well written; the appendix provides a nice mathematical background/review, many interesting remarks/extensions/experiments. The main strength of the paper arises from the observation that low rank matrices can accurately approximate gradients (where fidelity is measured by residual frobenius norm error). This approximation allows the proposed method to be essentially a factor of O(n) faster than its competitors.

One potential downside is that the proposed method (projUNN) and some of the baselines use eigen decomposition/QR decompositions which suffer some overhead due to parallelization issues in the GPU, causing some methods to appear slower in the experiments. Authors mention that this issue could be potentially fixed in the future with updates to DL libraries.

---

> ### Author Response · Authors · 2022-08-01
> **Response to reviewer BcYz**
>
> We thank the reviewer for their feedback and nice comments about the presentation and relevance of our manuscript. We also thank the reviewer for pointing out the typo in pg. 23 which we have corrected.
>
> **Reviewer:** “It would be interesting to compare train/test loss evolution over iterations between projUNN and the other baselines (i.e. testing convergence in terms of iterations rather than time) as well as total time to achieve similar loss (as opposed to single iteration cost).”  \
> **Response:** For most tasks considered here, we benchmarked performance with respect to reported results. Most of these tasks were learned with RNNs whose matrices are small enough to efficiently perform operations like matrix inversion and SVD, and thus runtimes, in most of these settings, are roughly equivalent for projUNN and other models. Since our goal was to address poor runtime scalings of the prior methods, we did not directly compare runtimes for the relatively small models considered. Even with that caveat, comparisons can be made by comparing convergence times in our plots to those of previous models. For example expRNN [1] and scoRNN [2] report learning the copy task (T=2000) in about 2000 and 4000 iterations (times taken from plots in the respective papers) respectively whereas projUNN learns the same task in about 3000-5000 iterations depending on the rank approximation in our experiment. In general, we find that since we are approximating gradients at each step with low rank matrices, projUNN needs more iterations to learn but this overhead is roughly a small constant multiple of comparable models. Looking ahead, we would like to analyze potential areas where the scaling benefits of projUNN have practical applications.
>
>
> **References**\
> [1] Lezcano-Casado, M., & Martınez-Rubio, D. (2019, May). Cheap orthogonal constraints in neural networks: A simple parametrization of the orthogonal and unitary group. In International Conference on Machine Learning (pp. 3794-3803). PMLR.\
> [2] ​​Helfrich, K., Willmott, D., & Ye, Q. (2018, July). Orthogonal recurrent neural networks with scaled cayley transform. In International Conference on Machine Learning (pp. 1969-1978). PMLR.

---

> > ### Comment · Reviewer_BcYz · 2022-08-08
> > **Thank you**
> >
> > Thank you for the detailed responses. Indeed, it will be interesting to analyze and identify potential practical applications.

---

### Meta-Review · Area_Chair_dbhX · 2022-08-21

**Recommendation:** Accept
**Confidence:** Certain

**Metareview:**

This paper provides two routines to replace gradient updates with low-rank unitary updates, and provides extensive technical discussion and experiments.  Reviewers are uniformly positive, and I also voice similar praises, e.g., I too appreciate the extensive discussion in appendices A and B, and the detailed experiments in the later appendices.  As such, it is easy to recommend acceptance, and I will push for this to receive at least a spotlight.  Even so, I urge the authors to make careful revisions for remaining issues raised by the reviewers, and to perform a full pass of their own.

**Award:**

No

---

### Decision · Program_Chairs · 2022-09-14

Accept